# The "Hydro-ABC model" (Vn 2.0): a simplified convective-scale model with moist dynamics

Jiangshan Zhu[1,*] and Ross Noel Bannister[2,*]

[1]Institute of Atmospheric Physics, Chinese Academy of Sciences, Beijing, P.R. China
[2]National Centre for Earth Observation, University of Reading, Reading, UK
[*]These authors contributed equally to this work.

**Correspondence:** Ross Bannister (r.n.bannister@reading.ac.uk)

**Abstract.** The prediction of convection (in terms of position, timing, and strength) is important to achieve for high-resolution weather forecasting. This problem not only requires good convective-scale models, but also data assimilation systems that give initial conditions which neither improperly hinder nor hasten convection in the ensuing forecasts. Solving this problem is difficult and expensive using operational-scale numerical weather prediction systems, and so a simplified model of convective-scale flow is under development (called the "ABC model"). This paper extends the existing ABC model of dry convective-scale flow to include mixing ratios of vapour and condensate phases of water. The revised model is called "Hydro-ABC".

Hydro-ABC includes transport of the vapour and condensate mixing ratios within a dynamical core, and transitions between these two phases via a micro-physics scheme. A saturated mixing ratio is derived from model quantities, which helps determine whether evaporation or condensation happens. Latent heat is exchanged with the buoyancy variable (ABC's potential temperature-like variable) in such a way to conserve total energy, where total energy is the sum of dry energy and latent heat. The model equations are designed to conserve the domain-total mass, water, and energy.

An example numerical model integration is performed and analysed, which shows the development of a realistic looking anvil cloud, and excitation of inertio-gravity and acoustic modes over a wide range of frequencies. This behaviour means that Hydro-ABC is a sufficiently challenging model which will allow experimentation with innovative data assimilation strategies in future work. An ensemble of Hydro-ABC integrations is performed in order to study the possible forecast error covariance statistics (knowledge of which is necessary for data assimilation). These show patterns that are dependent on the presence of convective activity (at any model's vertical column), thus giving a taste of flow-dependent error statistics. Candidate indicators/harbingers of convection are also evaluated (namely relative humidity, hydrostatic imbalance, horizontal divergence, convective available potential energy, convective inhibition, vertical wind, and condensate mixing ratio), some of which appear to be reliable diagnostics concerning the presence of convection. These diagnostics will be useful in the selection of the relevant forecast error covariance statistics when data assimilation for Hydro-ABC will be developed.

## 1 Introduction

Numerical models used for modern numerical weather prediction (NWP) services have evolved rapidly in sophistication and scale over recent decades. One of the most notable advances concerns the reduction in size of the models' grid boxes. Using

smaller grid sizes not only allows for finer flows to be resolved, but also permits new flow regimes, which could not be represented with coarser models (Clark et al., 2016; Yano et al., 2018). These flow regimes are more non-linear and inherently less predictable than larger-scale flows (Hohenegger and Schär, 2007; Leung et al., 2019; Lorenz, 1969) and thus there is an increased need for more observations to constantly correct for fast growing errors in order to produce useful forecasts (e.g. Banos et al. (2022)). This is not just a dynamical effect (for instance due to larger Rossby or Reynolds numbers), but is also due to the role of phase transitions of water (namely evaporation and condensation, thus absorbing or releasing latent heat (Errico et al., 2007)), which can vary over short length-scales, especially where moist processes are occurring (Montmerle and Berre, 2010).

By resolving moist processes that lead to convective motion (partially achieved with km-scale grid lengths), some aspects of numerical models have become simpler, as much of the fine-scale transport of water and latent heat can be handled explicitly, thus removing the need for a convection scheme. Conversely, the necessary techniques needed to realistically estimate the initial conditions (namely data assimilation, DA) have become more complicated owing to the breakdown in many of the assumptions that are made in traditional DA methods, like linearity of the models and observation operators, Gaussianity of the background and observation errors, and the homogeneity and quasi-static nature of the background error statistics (Bannister et al., 2020). These (and other) problems contribute to the difficulty of assimilating data (especially radar data) in such models (Fabry and Meunier, 2020).

High resolution models of the atmosphere and their DA counterparts remain very expensive to run, requiring costly super-computers. It is therefore desirable to study the convective-scale DA problem in simplified (or 'toy') models that are capable of being run at less cost on smaller computers. One of the first such simplified model was presented by Würsch and Craig (2014). Their model is based on the non-rotating 1D shallow water equations, comprising an equation each for fluid velocity, $u$, fluid depth, $h$, and rain water, $r$. A further variable, the geopotential, $\phi$, is tied to $h$ in a non-trivial way. When the total height, $H + h$ (where $H$ is the topography height) is less than a specified level, the system is considered sub-saturated and $\phi$ and $h$ are related via $\phi = g(H + h)$ (where $g$ is the acceleration due to gravity). When $h$ is above the specified level, a cloud is assumed to form and an analogue of a 'low pressure' region is formed by redefining $\phi = gH + \phi_c$ (where $\phi_c$ has a small value). This step encourages mass convergence and a form of convection to be initiated. Rain is then formed when $h$ is above a second specified level in regions of mass convergence. The model exhibits convection-like behaviour showing the sporadic appearance of clouds. Such a model, and its developments (Kent et al., 2017), provide useful first steps to provide a cheap and effective model on which to base developments in convective-scale data assimilation.

The ABC model (Petrie et al., 2017) was separately developed as a low-cost model of convective-scale behaviour also to be used as a basis to investigate convective DA methods. The ABC model is based on the three-dimensional Euler equations, but is reduced to two-dimensions (longitude and height). It is therefore more complex than the shallow water-based model of Würsch and Craig (2014), most notably in that it allows a height dependence. A variational DA capability was later developed for the ABC model (Bannister, 2020). This has since been used to show that enforcing geostrophic and hydrostatic balances as part of the background error covariance matrix (a key ingredient of the traditional variational DA problem) affects the DA's ability to analyse observations (Bannister, 2021). That study found that these balances, in the mid-latitude cases, are an advantage when

analysing the larger scales but are a disadvantage at small scales. The model has also been used in a tropical setting, where the DA method has been expanded to include a hybrid (ensemble-variational) capability, Lee et al. (2022). Until now however, the ABC model lacks a water variable, meaning that studies have been limited to dry dynamics only, where the model shows little evidence of sporadic convective behaviour. In this paper we show how moist processes can be included in the ABC model and how such behaviour emerges, which is a more appropriate setting for DA schemes to be tested and developed.

## 2 "Dry ABC" to "Hydro-ABC"


We seek modifications to the original ABC equations (Petrie et al., 2017) which are as simple as possible and requiring minimal change of the underlying numerical advection scheme. We refer to this moist model as "Hydro-ABC". The ABC model comprises five variables on a two-dimensional (longitude/height) grid: zonal wind $u$, meridional wind $v$, vertical wind $w$, scaled density perturbation $\tilde{\rho}'$ (a pressure-like variable), and buoyancy perturbation $b'$ (a potential temperature-like variable).

Hydro-ABC contains two new variables, namely water vapour $q$, and condensate $q_c$ mixing ratios, the latter of which may also be thought of a cloud variable. The model allows exchange between these two species according to the local thermodynamic conditions, with an associated latent heat exchange (see below). In order to keep the model as simple as possible, no rain is currently represented. The continuous dry equations support mass and total energy conservation, as shown in Petrie et al. (2017). Hydro-ABC must maintain similar conservation properties, but with the addition of conservation of total water, $q + q_c$,

and a total energy that includes a contribution from latent heat.

### 2.1 The Hydro-ABC model equations

The continuous Hydro-ABC equations are as follows:

$$\frac{\partial u}{\partial t} + B\boldsymbol{u} \cdot \nabla u + C\frac{\partial \tilde{\rho}'}{\partial x} - fv = 0, \tag{1a}$$

$$\frac{\partial v}{\partial t} + B\boldsymbol{u} \cdot \nabla v + fu = 0, \tag{1b}$$

$$\frac{\partial w}{\partial t} + B\boldsymbol{u} \cdot \nabla w + C\frac{\partial \tilde{\rho}'}{\partial z} - b' = 0, \tag{1c}$$

$$\frac{\partial \tilde{\rho}'}{\partial t} + B\nabla \cdot (\tilde{\rho}\boldsymbol{u}) = 0, \tag{1d}$$

$$\frac{\partial b'}{\partial t} + B\boldsymbol{u} \cdot \nabla b' + A^2 w = \mathcal{S}_{b'}, \tag{1e}$$

$$p' = C\rho_0 \tilde{\rho}', \tag{1f}$$

$$\frac{\partial \tilde{\rho}q}{\partial t} + B\nabla \cdot (\tilde{\rho}q\boldsymbol{u}) = \tilde{\rho}(Ev - Co), \tag{1g}$$

$$\frac{\partial \tilde{\rho}q_c}{\partial t} + B\nabla \cdot (\tilde{\rho}q_c\boldsymbol{u}) = -\tilde{\rho}(Ev - Co). \tag{1h}$$

Here $A$ is the pure gravity wave frequency, $B$ is the modulation of the advection and mass divergence terms, $C$ is the inverse compressibility coefficient, $t$ is time, $x$ and $z$ are positions in the longitude and height directions respectively, $\boldsymbol{u} = (u, v, w)$, $f$ is the Coriolis parameter, $\rho_0$ is a reference density, $\tilde{\rho}$ is the scaled density ($\tilde{\rho} = (\rho_0 + \rho')/\rho_0$), $p'$ is the pressure perturbation

(diagnosed purely from $\tilde{\rho}'$), $\mathcal{S}_{b'}$ is the buoyancy source term (related to latent heat exchange, which is activated when there is a change in water phase), $Ev$ is the evaporation rate, $Co$ is the condensation rate, and $\nabla = \left( \partial/\partial x \quad \partial/\partial z \right)$ is the two-dimensional vector derivative. The key differences between Hydro-ABC the dry ABC system are $\mathcal{S}_{b'}$ in (1e), and the additional equations (1g) and (1h). How $\mathcal{S}_{b'}$ and $Ev - Co$ are parametrised in Hydro-ABC is described below.

## 2.2 Conservation properties of the continuous equations

Petrie et al. (2017) (their Appendix B) uses Gauss' divergence theorem, the model's vertical boundary conditions (their Table 2), and periodic lateral boundary conditions, to show that the dry equations conserve total mass, $\int \int dx dz\, \rho$, and total dry energy, $\int \int dx dz\, E_{\mathrm{dry}}$, where the integration is done over the whole domain ($E_{\mathrm{dry}}$ is defined in Sect. 2.2.2). Here we show that the Hydro-ABC equations (1) also conserve total water, and total moist energy.

### 2.2.1 Conservation of total water

Total water at a particular location, $q_{\mathrm{t}}$, is the sum of the vapour and condensate mixing ratios, $q_{\mathrm{t}} = q + q_{\mathrm{c}}$. Adding (1g) and (1h) gives the governing equation for $q_{\mathrm{t}}$: $\partial(\tilde{\rho} q_{\mathrm{t}})/\partial t + B\nabla \cdot (\tilde{\rho} q_{\mathrm{t}} \boldsymbol{u}) = 0$. Multiplying this equation by the constant reference density $\rho_0$, recognising that $\rho = \rho_0 \tilde{\rho}$, and integrating over the domain gives the following equation for the total amount of water in the domain:

$$\frac{\partial}{\partial t} \int \int \rho q_{\mathrm{t}} dx dz + B \int \int \nabla \cdot (\rho q_{\mathrm{t}} \boldsymbol{u})\, dx dz = 0. \tag{2}$$

By following the same procedure to that in Petrie et al. (2017) (their Appendix B), it is straightforward to show that the second term is zero, thus demonstrating that the total amount of water in the domain is constant in time.

### 2.2.2 Conservation of total moist energy

From Petrie et al. (2017) the kinetic energy is $E_{\mathrm{k}} = \tilde{\rho}(u^2 + v^2 + w^2)/2$, the buoyancy energy is $E_{\mathrm{b}} = \tilde{\rho}b'^2/(2A^2)$, and the elastic energy is $E_{\mathrm{e}} = C\tilde{\rho}'^2/(2B)$. These make up the total dry energy, $E_{\mathrm{dry}} = E_{\mathrm{k}} + E_{\mathrm{b}} + E_{\mathrm{e}}$. In Hydro-ABC there is an additional latent heat energy which we define as

$$E_{\mathrm{l}} = \tilde{\rho} L_{\mathrm{v}} q, \tag{3}$$

where $L_{\mathrm{v}}$ is the specific latent heat of vapourisation of water. In Hydro-ABC, the total moist energy is defined as the sum of the dry energy and latent heat, $E_{\mathrm{moist}} = E_{\mathrm{dry}} + E_{\mathrm{l}} = E_{\mathrm{k}} + E_{\mathrm{b}} + E_{\mathrm{e}} + E_{\mathrm{l}}$.

In its continuous form, Hydro-ABC is designed to conserve $\int \int dx dz\, E_{\mathrm{moist}}$ (see Appendix A). We, however, solve the Hydro-ABC equations (1) over a timestep $\Delta t$ in two parts: a 'dynamical core' in which $\mathcal{S}_{b'}$, $Ev$, and $Co$ are each zero, followed by a micro-physics step. A similar time-step splitting is done in other models, e.g. WRF (Weather Research and Forecasting model, Skamarock et al. (2021)). The dynamical core is equivalent to integrating the dry equations (with extra

advection equations, for $q$ and $q_c$). The evolution of the total energy is represented by the following sum

$$\frac{\partial E_{\text{moist}}}{\partial t} = \underbrace{\left(\frac{\partial E_{\text{dry}}}{\partial t}\right)^{\text{dc}} + \left(\frac{\partial E_{\text{l}}}{\partial t}\right)^{\text{dc}}}_{(\partial E_{\text{moist}}/\partial t)^{\text{dc}}} + \underbrace{\left(\frac{\partial E_{\text{l}}}{\partial t}\right)^{\text{mp}} + \left(\frac{\partial E_{\text{b}}}{\partial t}\right)^{\text{mp}}}_{(\partial E_{\text{moist}}/\partial t)^{\text{mp}}}, \tag{4}$$

where the large brackets indicate that the changes are performed over the dynamical core (dc) or the micro-physics (mp).

The first term represents point-wise changes of the dry energy by the dynamical core; this is the part that is shown in Petrie et al. (2017) to integrate to zero over the model's domain. The second term represents point-wise changes of the latent heat energy by the dynamical core, which is driven by transport of $q$. Although $q$ (and hence $E_{\text{l}}$) can change at a particular point, the dynamical core does not change the phase of water, so this term also integrates to zero over the model's domain. The third and fourth terms are each non-zero, but the micro-physics scheme is designed so that their point-wise sum is zero (see Sect.

2.3). Hence integrating $\partial E_{\text{moist}}/\partial t$ over the entire domain is zero. Note that the reason why it is the buoyancy energy that appears in the last term in the micro-physics step in Eq. (4) (and not kinetic or elastic energies) is explained below. In Sect. 3.2 we examine how well such energy conservation is approximated in the time/space discretised equations.

### 2.3   The parametrisations of $\mathcal{S}_{b'}$, $Ev$ and $Co$

As explained above, Hydro-ABC solves equations (1) over a timestep $\Delta t$ in two stages, by first assuming $\mathcal{S}_{b'}$ and $Ev - Co$

are each zero (the dynamical core), and then their effects are added in a separate step (micro-physics, described below). The point-wise conservation of energy for the micro-physics step, $\Delta^{\text{mp}} E_{\text{moist}} = 0$ (where $\Delta^{\text{mp}} \bullet \approx (\partial \bullet /\partial t)^{\text{mp}} \Delta t$), is the basis for our micro-physics parametrisation, which is now described. The purpose of the micro-physics step is to determine values of $\mathcal{S}_{b'}$, and $Ev - Co$. There are no general closed-form expressions for these quantities, so some equations need to be solved implicitly, which are established in this section.

The relationship between the micro-physical change in $q$ and $Ev - Co$ is as follows: $\Delta^{\text{mp}} q = (Ev - Co) \Delta t$. Further, the relationship between the change in $E_{\text{l}}$ and the change in $q$ is: $\Delta^{\text{mp}} E_{\text{l}} = \tilde{\rho} L_{\text{v}} \Delta^{\text{mp}} q$. The above constraint that $\Delta^{\text{mp}} E_{\text{moist}} = 0$ may be put in the following way, $\Delta^{\text{mp}} E_{\text{moist}} = \Delta^{\text{mp}} E_{\text{k}} + \Delta^{\text{mp}} E_{\text{b}} + \Delta^{\text{mp}} E_{\text{e}} + \Delta^{\text{mp}} E_{\text{l}} = 0$. To facilitate closure, we assume that buoyancy energy is the only form of the dry energy that is exchanged with the latent heat. This is a reasonable assumption because in ABC, buoyancy is a potential temperature-like variable (hence only $E_{\text{b}}$ appears in the micro-physics terms of (4)).

The last equation then simplifies to $\Delta^{\text{mp}} E_{\text{b}} + \Delta^{\text{mp}} E_{\text{l}} = 0$. Combining the above formulae with the form of $E_{\text{b}}$ (which is given in the opening paragraph of Sect. 2.2.2) yields

$$\begin{aligned}
\frac{\tilde{\rho}}{2A^2} \Delta^{\text{mp}} \left(b'^2\right) &= -\tilde{\rho} L_{\text{v}} \left(Ev - Co\right) \Delta t, \\
\text{or } \Delta^{\text{mp}} \left(b'^2\right) &= -2A^2 L_{\text{v}} \left(Ev - Co\right) \Delta t.
\end{aligned} \tag{5}$$

Note that $\Delta^{\text{mp}} b' = \mathcal{S}_{b'} \Delta t$. Equation (5) will now be used with further rules (below) to determine how our micro-physics

scheme will modify $q$, $q_c$, and $b'$ (which will reveal how $\mathcal{S}_{b'}$ and $Ev - Co$ are found). The description of the scheme may be read in combination with the pseudo code presented in Appendix B.

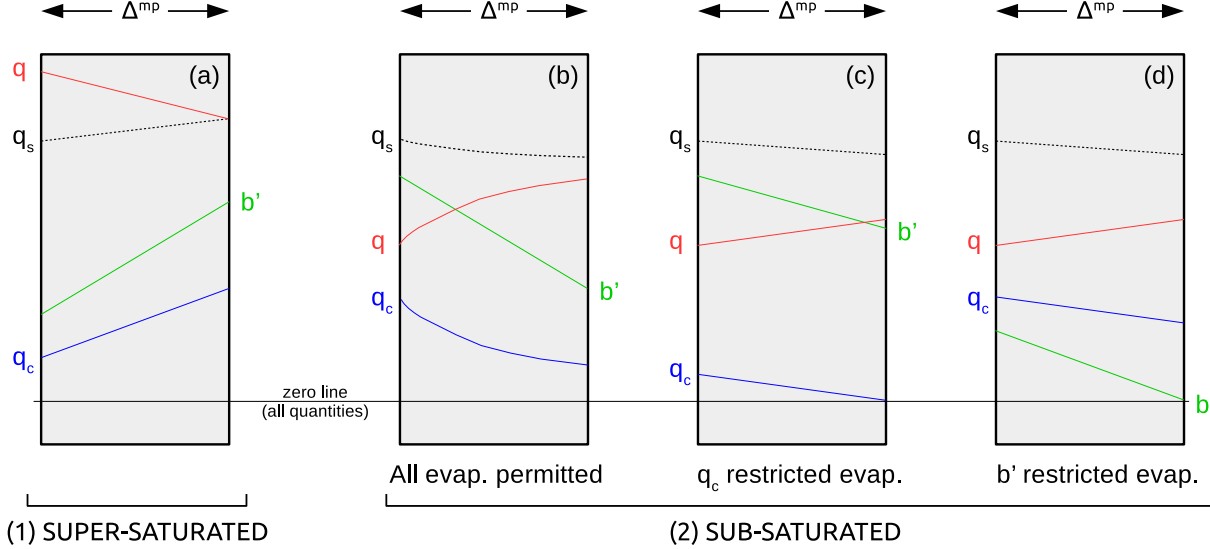

**Figure 1.** Graphical representation of Eq. (6), showing the change in $q$, $q_c$, and $b'$ over a micro-physics step. Water vapour mixing ratio is shown in red, condensate is in blue, and buoyancy is in green. The water vapour mixing ratio according to saturation is the black dotted line. The zero line for all quantities spans all panels. Case (1) concerns condensation of water vapour due to super-saturation and case (2) concerns evaporation of condensate into vapour where there are three possibilities. Case (3) is not shown, as it represents no micro-physical changes to the fields.

The task now is to determine $(Ev - Co)\,\Delta t$, which is the change in $q$ (or minus the change in $q_c$) over the micro-physics step. The following descriptions are made with reference to steps in the pseudo code in Appendix B (in the form of Ap-B, step $x$). Consider the following change in vapour over the micro-physics step, which depends on whether $q$ is super- or sub-saturated 150 after the dynamical core is run for one timestep:

$$(Ev - Co)\,\Delta t = \begin{cases} -(q - q_s) & \begin{aligned} &\text{Condensation (super-saturated):} \\ &\quad \text{when } q \geq q_s \text{ and} \\ &\quad (b' > 0 \text{ or } [b' < 0 \text{ and } \tilde{\rho} L_v\,|q - q_s|\,/E_b > \Gamma]) \end{aligned} \\ \min\left((q_s - q)\left(1 - e^{-\Delta t/\tau}\right), q_c, b'^2/\left(2A^2 L_v\right)\right) & \begin{aligned} &\text{Evaporation (sub-saturated):} \\ &\quad \text{when } q < q_s \text{ and } b' > 0 \end{aligned} \\ 0 & \text{Otherwise,} \end{cases} \tag{6}$$

where $q_s$ is the saturated mixing ratio (this is a function of the model state, see Sect. 2.4), $\Gamma$ is a threshold for convection (see below), $\tau$ is the prescribed evaporation timescale, and the "min" function selects the smallest of its three arguments. This equation is represented in Fig. 1. The three cases of this equation are now discussed.

1. The first case in Eq. (6) concerns condensation of water vapour due to super-saturation (Fig. 1a and Ap-B, step 3). In this case $q$ will instantaneously decrease by the amount $q - q_s$ (red line), and $q_c$ will increase by the same amount (blue line). According to (5), associated with this step is an increase in $b'^2$ by the amount $\Delta^{mp}\left(b'^2\right) = 2A^2 L_v (q - q_s)$. This is due to the release of latent heat, which is converted to buoyancy energy. Writing $\Delta^{mp}\left(b'^2\right)$ as $b'^2_{mp} - b'^2$ (where $b'_{mp}$ is the value of buoyancy after micro-physics, and $b'$ before), the equation for $\Delta^{mp}\left(b'^2\right)$ translates to

$$b'_{mp} = +\sqrt{b'^2 + 2A^2 L_v (q - q_s)} \tag{7}$$

(green line). The positive square-root is assumed because $b'$ is associated with potential temperature, which would rise after latent heat is released.

(a) Note that $q_s$ will be shown in Sect. 2.4 to be a function of $\tilde{\rho}'$ and $b'$. Changing $b'$ as prescribed by (7) will therefore change $q_s$. Equation (7) therefore represents a non-linear equation. This is solved by finding the root of the function

$f(q_{mp}) = q_{mp} - q_s(\tilde{\rho}', b'_{mp}(q_{mp}))$ given that $b'_{mp}$ is a function of $q_{mp}$, i.e. $b'_{mp} = +\sqrt{b'^2 + 2A^2 L_v (q - q_{mp})}$, and assuming that $\tilde{\rho}'$ is constant over the micro-physics step. When $b' \geq 0$ this is done using a Newton-Raphson procedure while assuming $\tilde{\rho}'$ is constant. The value of $(Ev - Co)\Delta t$ for the first branch of (6) is then $-\left(q - q_s(\tilde{\rho}', b'_{mp}(q_{mp}))\right)$. See Ap-B, step 3a.

(b) Step (a), however, becomes potentially unphysical if $b'$ is large and negative, and $q - q_s$ is small, as this would flip

the sign of $b'$ to be positive, resulting is an unrealistically large increment in $b'$. For this reason, included in the first case is the additional condition that $\tilde{\rho} L_v |q - q_s| / E_b > \Gamma$, where $\Gamma \gg 1$. This is the condition (which must be met only when $b' < 0$) that, for condensation to happen, the release of latent heat must be much larger than the amount of buoyancy energy. The value of $\Gamma = 10$ is chosen for the results in this paper. Also, for simplicity, when $b' < 0$, the amount of latent heat released is based on the value of $q_s$ at the pre-micro-physics value of $b'$, and not

on the value of $q_s$ at the value of increased value of $b'$ due to the latent heat release. This means that the non-linear equation in (a) above is not used in this case. See Ap-B, step 3b.

The cause of this issue seems to be in the square (of $b'$) and square-root in Eq. (7), which forces the sign of $b'$ to become invisible, followed by the need to choose the sign of the solution.

2. The second case in Eq. (6) concerns evaporation of condensate into vapour (see Ap-B, step 4), where $q$ is assumed

to relax exponentially upwards towards $q_s$ with timescale $\tau$ (Fig. 1b, red line), and $q_c$ relaxes downwards (blue line). According to (5), associated with this step is an change in $b'^2$ by the amount $\Delta^{mp}\left(b'^2\right) = -2A^2 L_v (Ev - Co)\Delta t$ (where $Ev - Co > 0$). This translates to a new value $b'_{mp}$ according to the following:

$$b'_{mp} = +\sqrt{b'^2 - 2A^2 L_v (Ev - Co)\Delta t}, \tag{8}$$

where the positive square-root is taken, which corresponds to a reduction in $b'$ due to evaporation (green line). This

process is restricted to when $b' > 0$, because in the case of $b' < 0$, (8) dictates that $b'$ will increase (and will instantly

become positive). Even if one chooses to take the negative square-root in cases when $b' < 0$, (8) will yield an increased $b'$ (i.e. $b'_{mp} - b' > 0$), which is unphysical.

This process assumes that there is an adequate amount of condensate to evaporate, and an adequate amount of buoyancy energy to convert to latent heat during evaporation (this corresponds to the first argument of the "min" function in (6) and to Ap-B, step 4a). The evaporation process is limited should either of these requirements not be met, which corresponds to the second and third arguments of the "min" function. When the second argument is smallest, evaporation is limited by available $q_c$ (panel c, where the micro-physics step evaporates all available $q_c$, and see Ap-B, step 4b), and when the third argument is smallest, evaporation is limited by available buoyancy energy (panel d, where the micro-physics step converts all available $E_b$, and see Ap-B, step 4c). As evaporation is much slower than condensation, $(Ev - Co)\Delta t$ (case 2 in (6)) is based on the value of $q_s$ before evaporation lowers the value of $b'$ (and hence lowers $q_s$). This avoids the need to solve the non-linear equations mentioned previously (but overall non-linearity is not avoided due the presence of the 'switches' in (6)).

3. The third case is invoked for the residual cases (i.e. for super-saturation but when the threshold for condensation is not met, or for sub-saturation but when $b' < 0$). In this case there are no micro-physics-related changes to the fields.

The change in buoyancy due to micro-physics, $b'_{mp} - b'$, yields the value $\mathcal{S}_{b'}\Delta t$ in (1e), and the change in vapour, $q_{mp} - q$, gives the value $(Ev - Co)\Delta t$ in (1g) and (1h).

## 2.4 The saturated vapour mixing ratio

The saturated mixing ratio, $q_s$, specifies the upper limit for water vapour. In standard atmospheric models it is a function of pressure and temperature. We use the following formula:

$$q_s = 1000\,\mathrm{g kg}^{-1}\frac{380\,\mathrm{Pa}}{p}\exp\left(\frac{17.3(T - 273.2\,\mathrm{K})}{T - 35.9\,\mathrm{K}}\right), \tag{9}$$

which is based on Eq. (9.8) of Pielke (2002) (adapted so that pressure is in Pa, and mixing ratio is in $\mathrm{g kg}^{-1}$). In (9), $p$ and $T$ (and hence $q_s$) have an implicit dependence on $x$, $z$, and $t$. Neither $p$ nor $T$ are ABC variables, and so we now take steps to produce physically reasonable formulae for $p$ and $T$ from ABC variables.

For pressure, a reference density profile, $\rho_0(z)$, is required. In the ABC model (Petrie et al., 2017), $\rho_0$ is a constant, but for the purposes of computing $q_s$ we let this depend on $z$ according to $\rho_0(z) = \rho_{00}\exp(-z/H)$, where $\rho_{00}$ is the surface air density ($1.225\,\mathrm{kg m}^{-3}$) and $H$ is the scale height (9km). For pressure, $p = p_0 + p'$, where $p_0$ (the reference pressure) is in hydrostatic balance with $\rho_0$: $dp_0(z)/dz = -\rho_0(z)g$, where $g$ is the acceleration due to gravity. The solution to this equation, given $\rho_0(z)$ is $p_0(z) = p_{00} - H\rho_{00}g\left(1 - \exp(-z/H)\right)$, where $p_{00}$ is the reference surface pressure. The relationship between surface reference pressure and surface reference density is $p_{00} = H\rho_{00}g$, which is derived from the previous equation with the condition $p_0(z \to \infty) \to 0$. ABC's equation of state (1f) relates the ABC variable $\tilde{\rho}'$ to $p'$, and using the above form of $p_o(z)$ gives

$$p(x, z, t) = p_0(z) + p'(x, z, t) = p_{00} - H\rho_{00}g\left(1 - \exp(-z/H)\right) + C\rho_{00}\exp(-z/H)\tilde{\rho}'(x, z, t). \tag{10}$$

In Petrie et al. (2017), $b'$ is related to a potential temperature increment $\theta'$ via $\theta' = (\theta_R/g)b'$, where $\theta_R$ is a constant (value $273\,\mathrm{K}$)[1]. Hence we focus now on potential temperature. The potential temperature in Petrie et al. (2017) is defined as $\theta(x,z,t) = \theta_R + \theta_0(z) + \theta'(x,z,t)$. To derive a reference profile, $\theta_0(z)$, we use the definition of the Brunt-Väisälä frequency $N^2 = (g/\theta_R)d\theta_0(z)/dz$ (see e.g. Holton and Hakim (2013)). The parameter $A$ in (1e) takes the place of $N$, and so we solve the equation $A^2 = (g/\theta_R)d\theta_0(z)/dz$. The solution to this equation is $\theta_0(z) = \theta_{00} - \theta_R + (A^2\theta_R/g)z$, where $\theta_{00}$ is the surface potential temperature (300K). The potential temperature is therefore

$$\theta(x,z,t) = \theta_R + \theta_0(z) + \theta'(x,z,t) = \theta_{00} + \frac{\theta_R}{g}\left(A^2 z + b'(x,z,t)\right). \tag{11}$$

The standard relationship for $T$ in terms of $\theta$ and $p$ is $T(x,z,t) = \theta(x,z,t)\left(p(x,z,t)/p_{00}\right)^{\kappa}$ (Holton and Hakim, 2013), where $\kappa$ is the constant $R/c_p$ (where $R$ is the gas constant for dry air and $c_p$ is the specific heat capacity at constant pressure; value of $\kappa$ is 0.286). Using (11) and (10) gives

$$T(x,z,t) = \left[\theta_{00} + \frac{\theta_R}{g}\left(A^2 z + b'(x,z,t)\right)\right]\left[\frac{p_{00} - H\rho_{00}g\left(1 - \exp(-z/H)\right) + C\rho_{00}\exp(-z/H)\tilde{\rho}'(x,z,t)}{p_{00}}\right]^{\kappa}. \tag{12}$$

Equations (10) and (12) can now be substituted into (9) to yield a physically reasonable function for the saturated vapour mixing ratio in terms of ABC variables $b'$ and $\tilde{\rho}'$.

## 2.5 Numerical details

The ABC model stores variables on an Arakawa C grid in the horizontal and a Charney-Phillips grid in the vertical (see Petrie et al. (2017), Fig. 1). The two extra variables in Hydro-ABC, $q$ and $q_c$, are each stored at the same location as $\tilde{\rho}$ and $\tilde{\rho}'$. This choice makes sense from a numerical perspective as each of $q$ and $q_c$ is multiplied by $\tilde{\rho}$ in (1g) and (1h), and most of the finite differences that approximate the derivatives in these equations evaluate at the correct locations on the grid.

In Hydro-ABC, (1g) and (1h) are solved as advection/source equations rather than as the continuity equations shown. By employing the product rule for differentiation, (1g) becomes $\tilde{\rho}\partial q/\partial t + q\partial\tilde{\rho}/\partial t + Bq\nabla\cdot(\tilde{\rho}\boldsymbol{u}) + B\tilde{\rho}\boldsymbol{u}\cdot\nabla q = \tilde{\rho}(Ev - Co)$. The second and third terms sum to zero due to the mass continuity equation (1d), leaving the advection/source equation for $q$:

$$\frac{\partial q}{\partial t} + B\boldsymbol{u}\cdot\nabla q = Ev - Co. \tag{13}$$

A similar equation is also found for $q_c$.

The dynamical core's scheme is based on Cullen and Davies (1991) and has two parts: an adjustment stage and an advection stage. The adjustment stage operates over two sub-timesteps and deals with the momentum and thermodynamic equations (omitting the advective terms) followed by treatment of the continuity equation. The advection stage advects the fields using the winds found in the adjustment stage averaged over the two sub-timesteps. Upwind gradients are used for the advection. For more details see Sect. 3.3 of Petrie et al. (2017). The equations for $q$ and $q_c$ (with $Ev - Co = 0$) are each solved by

---

[1]This relationship between $b'$ and $\theta'$ does not imply that $b'$ has the full properties of potential temperature in the ABC model (e.g. that $b'$ is conserved during adiabatic processes). In Petrie et al. (2017) this relationship was used as a starting point in the derivation of the ABC equations from the Euler equations and it is re-used here as a pragmatic way of deriving temperature from the ABC variables.

means of the same advection scheme as for the other variables in ABC. $Ev - Co$ is subsequently found with the micro-physics parametrisation of Sect. 2.3 using values of variables after the dynamical core is called.

## 3 Example run of the model

In this section we show example behaviour of Hydro-ABC. The initial conditions for the dry variables ($u$, $v$, $w$, $\tilde{\rho}'$, and $b'$) are set according to the prescription in Petrie et al. (2017) (their Sect. 5.2). This involves extracting $u$ and $v$ from a longitude/height slice output from a $1.5\,\mathrm{km}$ Unified Model (UM) run, which has the same grid staggering as the ABC model, and then adjusting $u$ and $v$ to have smooth periodic boundary conditions in $x$. Then $\tilde{\rho}'$ is set using geostrophic balance, $b'$ is set using hydrostatic balance, and $w$ is set by imposing a 3-D divergence-free condition. The initial water vapour is set as shown in the top panel of Fig. 2 (set so the maximum relative humidity is 0.98), and the initial $q_c = 0$.

### 3.1 Reference run

The rest of Fig. 2 (rows 2, 3, and 4) shows the evolution of $w$ (left column) and $q$, $q_c$ (right column) over the first six hours of the integration using the reference parameters indicated in the caption. Before the first hour of the simulation has completed there are no obvious signs of convection (not shown). It is only at $t = 60\,\mathrm{min}$ that localised upward motion starts to develop at ground level most strongly at $x = 148.5\,\mathrm{km}$, but also at $156\,\mathrm{km}$ (also not shown). By $120\,\mathrm{min}$, these have developed into striking convective plumes, which transport vapour upwards. This coincides with the position of the water vapour maximum in the initial conditions. In this region there is significant conversion of vapour to condensate, thus releasing latent heat, which is converted to buoyancy energy, thus inducing strong upward motion. The magnitude of the plume's vertical motion is very high (peaking at $\sim 200\,\mathrm{ms}^{-1}$ at the strongest convecting point), but in the ABC model this speed is moderated by the small value of $B$ (0.05), so it is equivalent to a peak vertical mass transport of $10\,\mathrm{ms}^{-1}$, which is realistic. The vertical grid spacing is $\sim 250\,\mathrm{m}$, making the vertical Courant number 0.08 for the unmodulated motion and 0.004 allowing for the $B$-modulation. These values are well within the Courant–Friedrichs–Lewy condition for the Courant number to be less than unity.

By $240\,\mathrm{min}$, an anvil cloud of vapour and condensate has established itself between heights of $\sim 5.5$ to $8.5\,\mathrm{km}$, which spreads out laterally in both directions. The zonal winds at these levels peak at around $\pm 100\,\mathrm{ms}^{-1}$ (not shown), but again once $B$ moderation is taken into account this is equivalent to a horizontal mass transport of $\pm 5\,\mathrm{ms}^{-1}$. At this time, the second smaller plume has been extinguished by downdrafts from the main plume, but smaller convective plumes 11 to $13\,\mathrm{km}$ have developed on either side of the main plume. Between 240 and $360\,\mathrm{min}$, the smaller plumes are themselves impeded by down welling air on each side of the main plume. By $360\,\mathrm{min}$ the condensate anvils have continued to develop laterally, and the central plume has started the process of weakening.

### 3.2 Numerical effects on the total energy evolution

Hydro-ABC in its continuous form, as described by (1) and the algorithm in Sect. 2.3, will conserve total moist energy, $\int \int dx dz\, E_{\mathrm{moist}}(t) = \mathrm{constant}$. Discretisation of the equations in space and time will though lead to loss of energy conserva-

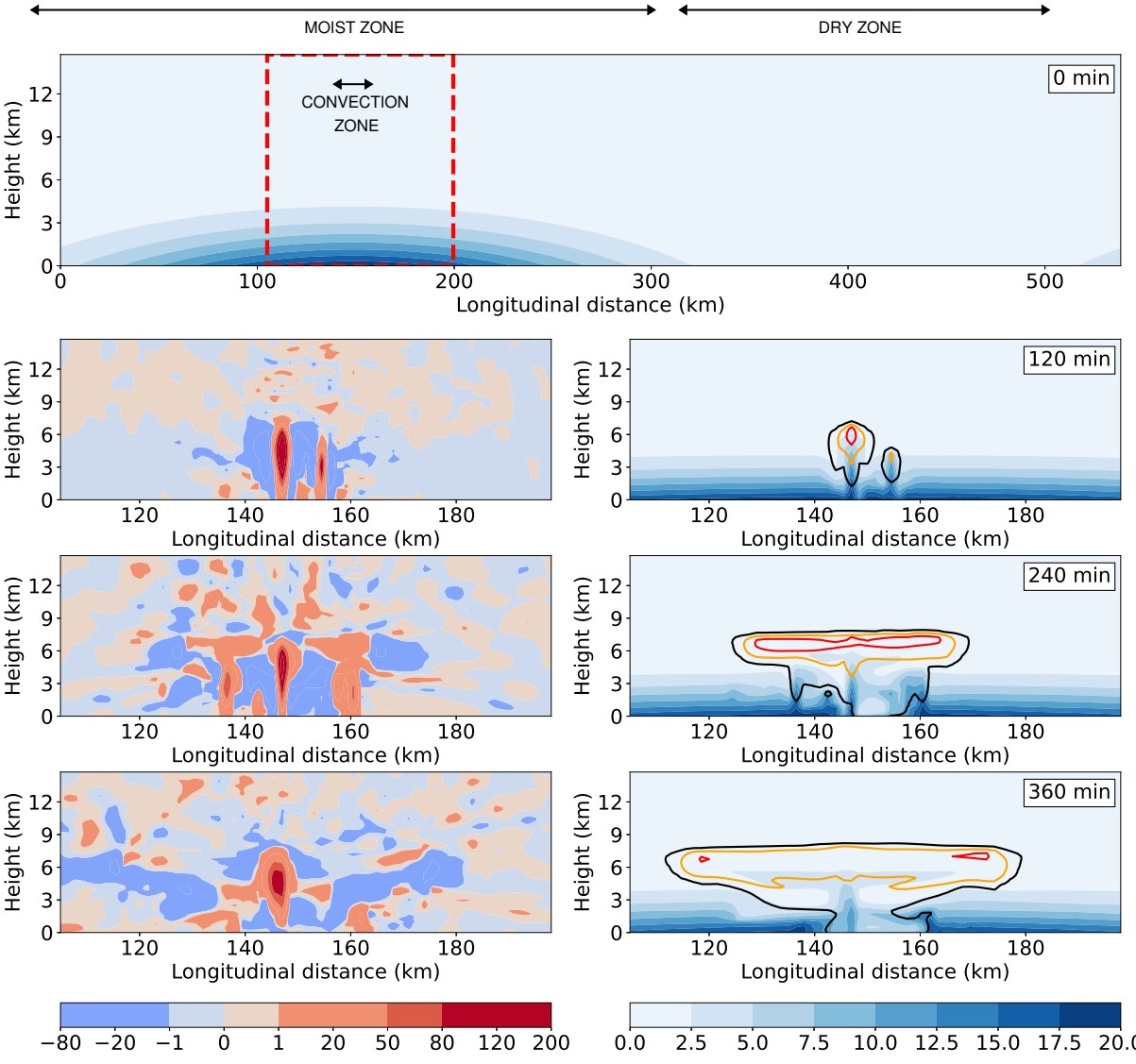

**Figure 2.** Initial conditions of the example reference run of Hydro-ABC ($A = 0.01\,\mathrm{s}^{-1}$, $B = 0.05$, $C = 2 \times 10^4\,\mathrm{m}^2\mathrm{s}^{-2}$, $f = 10^{-4}\,\mathrm{s}^{-1}$, $\Delta t = 0.1\,\mathrm{s}$, $\tau = 1000\,\mathrm{s}$, $L_\mathrm{v} = 2500\,\mathrm{J}\mathrm{g}^{-1}$, and $\Gamma = 10$). The initial conditions of the dry variables are set as described in the text, the initial values of $q$ are shown in the top panel (set so the maximum relative humidity is 98%), and the initial $q_\mathrm{c} = 0$. Snapshots of the subsequent integration of Hydro-ABC are shown in the remaining panels, zoomed into the region indicated by the red box in the top panel for: 120, 240, and 360 min. $w$ is shown in the left column ($\mathrm{ms}^{-1}$), and $q$ (shading) and $q_\mathrm{c}$ (contours) are shown in the right column. The contours are the black ($1\,\mathrm{g}\mathrm{kg}^{-1}$), orange ($6\,\mathrm{g}\mathrm{kg}^{-1}$) and red ($12\,\mathrm{g}\mathrm{kg}^{-1}$) lines. Three zones are defined, the "convection zone" (columns close to the convection), the "moist zone" (columns of non-zero moisture in the initial conditions), and the "dry zone" (zero moisture in the initial conditions).

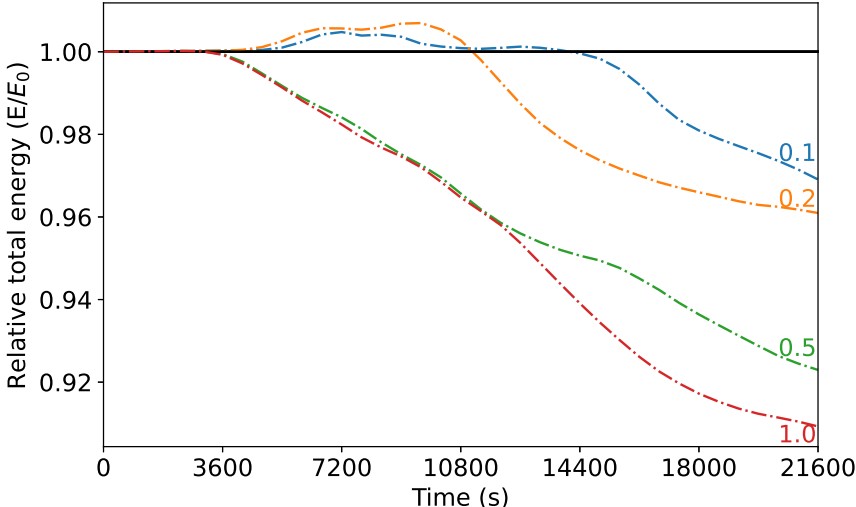

**Figure 3.** Evolution of the relative total energy of runs of hydro-ABC for different model integration timesteps as indicated (in s). The relative total energy is $\left[\int\int dxdz\, E_{\mathrm{moist}}(t)\right]/\left[\int\int dxdz\, E_{\mathrm{moist}}(0)\right]$, where $E_{\mathrm{moist}} = E_{\mathrm{k}} + E_{\mathrm{b}} + E_{\mathrm{e}} + E_{\mathrm{l}}$ (Sect. 2.2.2).

tion. Five numerical experiments – based on the reference parameters, but of different $\Delta t$ from $0.1$ to $1.0\,\mathrm{s}$ – are performed to analyse how in practice the total moist energy deviates from conservation. Figure 3 shows the evolution of relative total moist energy (relative to the total moist energy at $t = 0$), where the labels are the $\Delta t$ values of the model runs. All runs show
280    fluctuations in this quantity, which become visible on the plot's scale when convection starts just after $t = 60\,\mathrm{min}$. As expected $\Delta t = 0.1\,\mathrm{s}$ deviates from conservation the least, and $\Delta t = 1.0\,\mathrm{s}$ the most. The slight increase in energy in the $\Delta t = 0.1$ and $0.2\,\mathrm{s}$ integrations is small, but unexpected. Our assumption is that the $\Delta t = 0.1\,\mathrm{s}$ setting represents an acceptable deviation from conservation, especially given the fast-moving nature of the features that are being simulated. $\Delta t = 0.1\,\mathrm{s}$ is the setting that has been used for the result in Fig. 2, and for the remainder of this paper.

285    **3.3    Excitation of waves by convection**

The ABC model is capable of supporting Rossby-like, inertio-gravity and acoustic waves. The continuous equations can support all horizontal wavenumbers allowed by the periodic boundary conditions, and all frequencies. In practice though the maximum horizontal wavenumber and frequency are set by $\Delta x$ (the grid box length) and $\Delta t$ (the timestep) respectively. In runs of the dry model, high wavenumbers and frequencies have a very small weight, but their presence is still important in order to
290    represent scales that would otherwise anomalously dissipate energy. Allowing moist convection, as facilitated by Hydro-ABC, potentially excites appreciable weight to modes of high wavenumber and frequency.

Figure 4 shows $w$ for level 30 (near the top of the anvil cloud) of two model runs: one for the dry model (left) and another for Hydro-ABC (right). The top panels are Hovmöller plots of $w$ for the longitude/time domain. Both versions of the model show

westward and eastward propagating waves. The bottom panels of Fig. 4 show the spectral density of $w$ at the same height. These plots are simply the magnitude bi-Fourier transform of the top panels. This reveals the dispersion relations of the dominant waves in the system. By analogy with waves in systems that represent more closely the real atmosphere, each feature on these plots represents a wave with a different vertical wavenumber (Salby (1996), his Fig. 14.9) and the lines separate into two groups. The first group is the lower frequency inertio-gravity waves, whose frequencies increase with horizontal wavenumber and then level off at $A/(2\pi) = 1.59 \times 10^{-3}\,\mathrm{s}^{-1}$. This is the wave frequency of the highest frequency inertio-gravity waves in the high wavenumber limit (the horizontal dashed lines) which is consistent with classical wave theory, e.g. Gill (1982); Salby (1996). The second group is the higher frequency acoustic waves, whose frequencies increase with horizontal wavenumber. The quantity $\sqrt{BC}$ is the pure sound wave speed (the gradient of the sloping dashed lines). This lines separate the inertio-gravity waves and the acoustic waves and is occupied by Lamb waves in the real atmosphere (there is no visible evidence of Lamb waves in these (Hydro)-ABC runs, but we will still call this line the "Lamb line"). In the (Hydro)-ABC runs some of the high-wavenumber acoustic waves have frequencies lower than the Lamb line (visible especially in the top-right section of the bottom-right panel). This is a numerical effect where the frequencies of high-wavenumber waves are too low, leading to numerical dispersion (see e.g. Chapter 2 of Durran (1999)).

Comparing the dry ABC and Hydro-ABC spectra, the shapes of the dispersion curves look similar. This indicates that adding the adiabatic moisture scheme is not noticeably modifying the underlying dispersion relations of the dynamical core. Instead, the striking effects that moist convection in Hydro-ABC causes are (i) the excitation of stronger inertio-gravity and acoustic modes across the spectrum, and (ii) broadening of their frequencies. Effect (i) is visible at all wavenumbers and frequencies, especially for inertio-gravity waves for wavenumbers less than $\sim 70$. Effect (ii) is likely due to the gradual onset of convection, where the degree of broadening is inversely proportional to the timescale of the onset of convection. In the top-right panel of Fig. 4, there is a gradual onset of the strength of convection, which leads to a convolution of the spectra with this 'low frequency' onset.

## 4 Convection-dependent covariances and identification of areas prone to convection

It is well known that latent heat release can lead to convective instabilities. Such rapidly changing conditions present challenges for DA, not least because the background error covariances can become highly flow-dependent, e.g. Lorenc (2007); Montmerle and Berre (2010); Sun et al. (2014); Bannister et al. (2020). In this section we show how vertical error statistics of model variables depend on the background conditions for hydrodynamically stable and unstable regions. An important issue in convective-scale DA is the identification of convective regions, so that the (otherwise static) background error covariance matrix can be modified with the relevant vertical covariance statistics should the model be used with data assimilation.

### 4.1 How forecast statistics are affected by convectively-prone regions

In order to give a taste of moisture-affected covariances and correlations between different variables and positions, and in different regimes in hydro-ABC, we generate an ensemble of hydro-ABC forecasts. Longitude/height slices from a 93-member

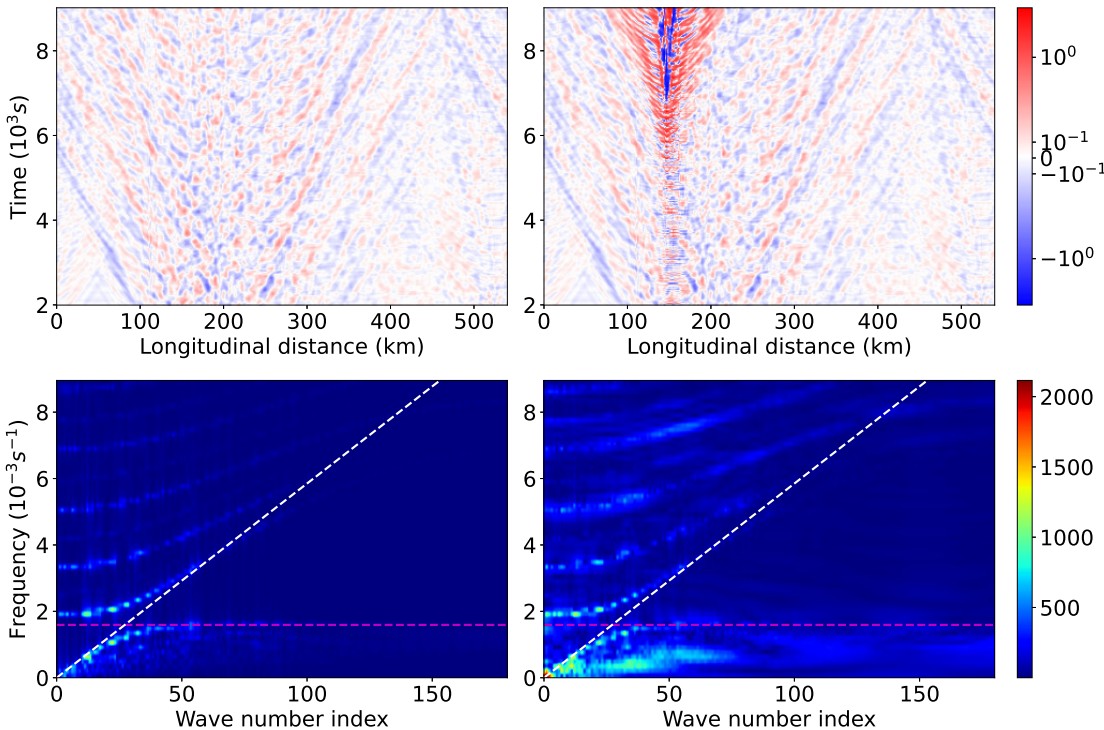

**Figure 4.** Vertical wind in the longitude/time (top panels) and wavenumber/frequency (bottom panels) domains for level 30 in the model (7.7 km height). The left panels are for dry ABC and the right panels are for hydro-ABC. The initial conditions of $u$, $v$, $w$, $\tilde{\rho}'$, and $b'$ are identical for ABC and Hydro-ABC, and the additional initial conditions for $q$ and $q_c$ are the same as those used in Fig. 2. In the bottom panels, the horizontal dashed lines are each at frequency $A/(2\pi)$ (the pure gravity wave frequency), and the sloped dashed lines each has gradient $df/dn = L^{-1}\sqrt{BC}$ (the pure sound wave speed), where $f$ is frequency (the $y$-axes), $n$ is the wavenumber index (the $x$-axes), and $L$ is the length of the domain, $540 \times 10^3$ m.

ensemble of convective-scale UM forecasts (Bannister et al., 2017) are each processed in the same way as described in Sect. 3 to give a 93-member ensemble of hydro-ABC forecasts (run out to 360 min). Each ensemble member is prepared with the same $q$ and $q_c$ fields (as in Fig. 2), but the relative humidity ($q/q_s$, the field that matters for convection) is different between members because $q_s$ is a function of $b'$ and $\tilde{\rho}'$ (see Eq. (9) Sect. 2.4, where $p$ and $T$ depend on $\tilde{\rho}'$ and $b'$ as shown in Eqs. (10) and (12)).

The example hydro-ABC run in Sect. 3 is one of these members, and convection happens around the same place in all but one member allowing the vertical statistics (covariances between different vertical levels) to be based on slightly different versions of the convective situation, which is centred at $x = 148.5$ km as in Fig. 2.

The covariances of and between different model variables will have different units and magnitudes. In order to simplify comparison of covariances, we first normalise each model variable by its (constant) maximum standard deviation, where the

maximum is calculated over all positions and times. This will result in covariances (referred to as normalised covariances) that

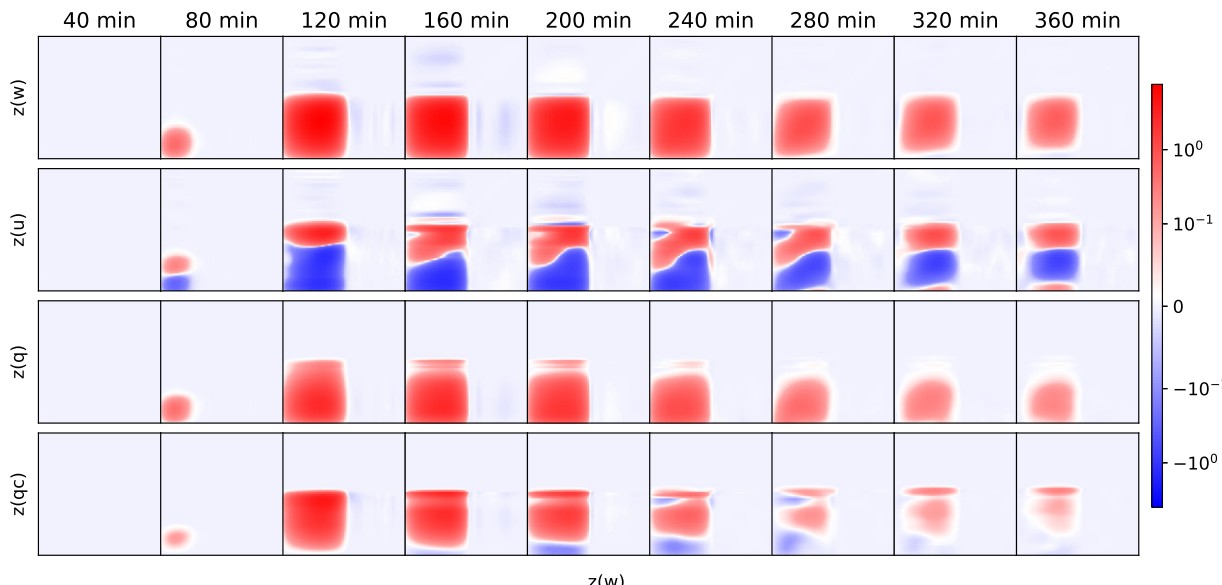

**Figure 5.** Evolution of a selection of normalised vertical covariances in the convection zone ($x = 148.5\,\mathrm{km}$) at times between $40$ and $360\,\mathrm{min}$. The rows are for $w$-$w$, $u$-$w$, $q$-$w$, and $q_\mathrm{c}$-$w$ covariances. In each plot of $p_1$-$p_2$ covariances, the $y$-axis is for $p_1$ and the $x$-axis is for $p_2$, and the surface is at the bottom left of each square. Red (blue) indicates positive (negative) covariances according to the logarithmic scale shown.

are bounded between $-1$ and $1$. These values are not correlations though as correlations are based on normalisation by the local standard deviations (local in space and time), although we do later also show correlations of different quantities.

Figure 5 shows the normalised covariances for a selection of quantities (see Fig. caption) at different times in the convection zone ($x = 148.5\,\mathrm{km}$). Near the start of the integrations ($t = 40\,\mathrm{min}$), all covariances shown are too small to be detected on
340 the scale used. At $80\,\mathrm{min}$, convection is just starting, which appears in a region of non-zero covariances within $4.5\,\mathrm{km}$ of the surface. There are positive covariances between $w$ and itself at different levels, and with $q$ (and more weakly with $q_\mathrm{c}$), and split negative/positive covariances with $u$. These results show that larger $w$ is associated with greater vertical advection of vapour, and the formation of a shear zone in zonal wind, with westward wind in the lower part and eastward wind in the upper part of this $4.5\,\mathrm{km}$ layer. By $120\,\mathrm{min}$ the non-zero covariances region has grown to within $9$-$10\,\mathrm{km}$ of the surface, including
345 growth of the wind shear zone. An increase in the vertical length-scales in $q$ errors is also seen in the WRF-based study of Michel et al. (2011) and in the AROME-based study of Montmerle and Berre (2010), in each case when the underlying conditions are raining (which is assumed to be comparable to the convective situation here where the condensate would fall as rain if precipitation processes were included). Additionally, $q_\mathrm{c}$ has developed strong positive covariances with $w$ in this region indicating that condensate has formed, with larger amounts with increasing $w$. As indicated in Fig. 2 (contours in right panels),
350 the condensate concentrations increase with height up to $\sim 9\,\mathrm{km}$. This suggests that the region of positive $q_\mathrm{c}$-$w$ covariances just mentioned is not caused by advection (the covariances would have negative sign if they were), but instead by the process

of more condensate being produced in association with stronger $w$. Note that in most parts of the cloud layer $q$ and $q_c$ are both positively correlated with $w$, which may appear counter-intuitive. In the convection zone, convergence and condensation processes are happening. The convergence can increase $q$ while the condensation will decrease $q$, but increase $q_c$. As long as
the increase of $q$ caused by convergence is larger than the loss of $q$ by condensation, there is a net increase in $q$, in which case both $q$ and $q_c$ could have a positive correlation with the vertical wind. In fact statistics show that often $q$ and $q_c$ are mutually positively correlated at points inside the cloud (not shown).

At 160 min a shallow surface layer of negative $q_c$-$w$ covariances has developed, which may be due to a circulation being induced by the convection, which is drawing in air with lower $q_c$ from the surroundings. There is an additional region of weak,
but non-zero, $w$-$w$ and $u$-$w$ correlations above the top of the convection, the latter indicating a further (but weak and shallow) shear region at the higher levels, which continues to 200 min. From 240 min the covariances develop more complicated, but weaker, patterns, including the development of a further possible $u$ shear layer near the surface. There is also an influence of the convection in vertical covariances at locations other than at $x = 148.5$ km, including at locations within and beyond the moist 'bubble' set-up in the initial conditions (covariances not shown, but correlations are shown below).

Figure 6 shows vertical correlations between a different selection of quantities to those in Fig. 5. The top three rows show auto-correlations of $\tilde{\rho}'$ for the convection, moist, and dry zones respectively (see Fig. 2 for the definition of the zones where the columns are selected, and Fig. 6 caption for their lateral positions). At $t = 40$ min the different zones show similar patterns. The model variable $\tilde{\rho}'$ appears to comprise two layers of correlations (one below 6 km, and the other above), where in each, $\tilde{\rho}'$ is strongly correlated vertically, but only weakly (and negatively) across each layer. In the convection zone there are some
small-scale vertical oscillations, which are not present in the other zones shown and so are likely to be due to early signs of convection. At 80 min in the convection zone, the lower layer level itself splits into two negatively correlated sub-layers, which are associated with the establishment of upper and lower pressure changes formed in connection with the $u$-$w$ covariances in Fig. 5 at this time (the wind shearing). The covariances in the moist and dry zones have not changed qualitatively very much at this time. At 120 and 160 min in the convection zone the sub-layering has disappeared (probably because the shear
zone has moved upwards), and the vertical lengthscales above 6 km have dramatically reduced. This shortening of the vertical lengthscales is also seen in the moist and dry zones, but with a delay of tens of minutes. From 200 min onwards the vertical covariances in the convection zone do not change very much in structure, but the other zones develop strong large-scale (vertically) correlation structures presumably due to outward propagation of disturbances from the convection zone.

The bottom three rows of Fig. 6 show vertical correlations between $b'$ at difference heights ($y$-axis) and $\tilde{\rho}'$ at different heights
($x$-axis). These are included in combination with the $\tilde{\rho}'$ auto-correlations to look for signatures of hydrostatic balance (HB). The HB equation is found from (1c) after the Lagrangian derivative terms are neglected, resulting in $C\partial\tilde{\rho}'/\partial z = b'$. This relates $b'$ to vertical gradients in $\tilde{\rho}'$. Thus the property of HB is assumed when the vertical derivative of the $\tilde{\rho}'$-$\tilde{\rho}'$ correlation function at a point in the top three rows of Fig. 6 is zero, and is matched by the $b'$-$\tilde{\rho}'$ correlation also being zero at the corresponding point in the bottom three panels. This property is seen at many points in Fig. 6 and a sample of points is highlighted in the Fig.
and below.

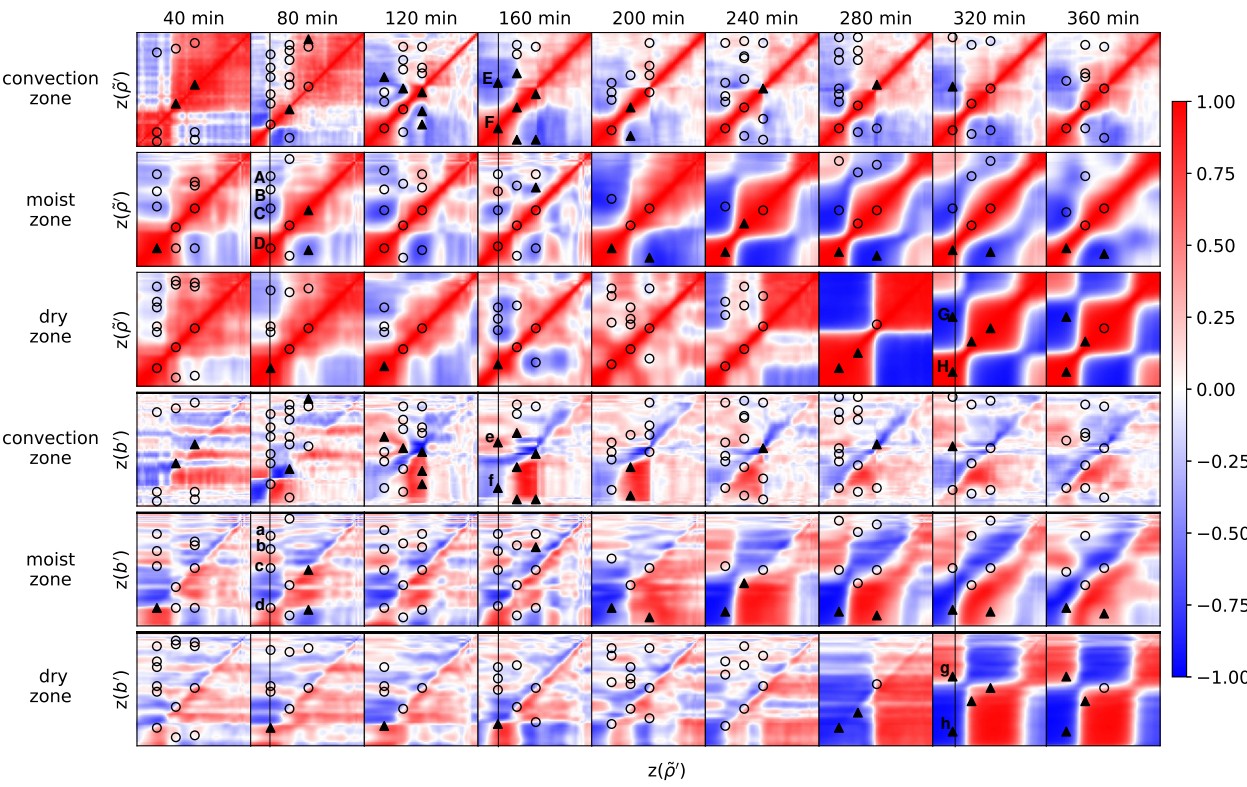

**Figure 6.** Evolution of a selection of vertical correlations at times between $40$ and $360\,\mathrm{min}$. The top set of three rows is for $\tilde{\rho}'$-$\tilde{\rho}'$ correlations, and the bottom set is for $b'$-$\tilde{\rho}'$ correlations. For each set of three rows, the rows are in the convection zone ($x = 148.5\,\mathrm{km}$ – the same position as used for Fig. 5), the moist zone ($286.5\,\mathrm{km}$), and the dry zone ($421.5\,\mathrm{km}$) respectively. In each plot of $p_1$-$p_2$ correlations, the $y$-axis is for $p_1$, the $x$-axis is for $p_2$, and the surface is at the bottom left of each square. Red (blue) indicates positive (negative) correlations. The circular and triangular symbols represent points where consistency with hydrostatic balance is tested. These are placed at a selection of heights in the top three rows where there is a local maximum or minimum in $\tilde{\rho}'$-$\tilde{\rho}'$ correlation with respect to the $y$-axis. Each symbol is also plotted in the corresponding bottom three rows to see if it associated with a 'small' $b'$-$\tilde{\rho}'$ correlation. Circles are associated with small $b'$-$\tilde{\rho}'$ correlations ($\leq 0.3$, associated with hydrostatic balance) and triangles otherwise. Some symbols are labelled with letters, which are referenced in the text.

In the top three rows, symbols are placed at a selection of points where the vertical correlation is stationary $(\partial(\tilde{\rho}'\text{-}\tilde{\rho}' \text{ correlation})/$ $\partial z(\tilde{\rho}') = 0)$. The corresponding symbols are placed at the same positions in the bottom three panels. A symbol is a circle if it has a small (proxy for zero) $b'\text{-}\tilde{\rho}'$ correlation magnitude ($\leq 0.3$), which is associated with HB, or a triangle otherwise. At $80\,\mathrm{min}$ for instance in the moist zone, there are four stationary points highlighted (A, B, C, and D). These correspond to near zero values of $b'\text{-}\tilde{\rho}'$ correlations (a, b, c, and d). There are only a few points in the moist and dry zones in the early stages of the evolution that do not satisfy this condition (triangles). At $160\,\mathrm{min}$ in the convection zone, there are two stationary points highlighted (E and F). These do not correspond to near zero values of $b'\text{-}\tilde{\rho}'$ correlations (e and f), suggesting a clear breakdown of HB. In the first half of the forecast period, there are more non-hydrostatic signatures in the convection zone than in the other zones. Later in the convection zone, the hydrostatic signatures return as the system stabilises. In the second half of the forecast period, the moist and then dry zones experience more non-hydrostatic signatures as a result of the outward propagation of disturbances from the convection zone. At $320\,\mathrm{min}$ in the dry zone for instance, there are two stationary points highlighted (G and H). These do not correspond to near zero values of $b'\text{-}\tilde{\rho}'$ correlations (g and h), suggesting a propagated breakdown of HB. Bannister et al. (2011) and Vetra-Carvalho et al. (2012) similarly found evidence of non-hydrostatic patterns in the covariance patterns in rainy regions (compared to dry regions) of forecast errors of the convective-scale UM. These statistics encourage the development of data assimilation systems that do not always constrain the background error covariances to be hydrostatic when convection is happening. Hydrostatic imbalance is further mentioned in Sect. 4.2.

### 4.2 Possible indicators and harbingers of convection

Since there are significant differences in the covariance and correlation structures for convecting and non-convecting regions, convective-scale data assimilation systems should gain advantages by being aware of the convection 'status' of any geographical region, and thus use appropriate background error statistics. This is done automatically in ensemble-based systems (e.g. ensemble Kalman filters or ensemble-variational methods, Bannister (2017)), and partially in hybrid systems (where the static background error covariances of traditional variational methods are averaged with ensemble-derived covariances, ibid.). It is well known though that the use of ensemble-based covariances leads to rank deficiency and under-sampling (Houtekamer and Mitchell, 1998; van Leeuwen, 1999; Hamill et al., 2001; Houtekamer and Mitchell, 2005; Ehrendorfer, 2007; Zhang and Zhang, 2012; Houtekamer and Zhang, 2016), and so any progress that allows the (otherwise) static background error covariance matrix to gain flow dependence is still valuable. This is the strategy behind a number of studies, including Montmerle and Berre (2010); Ménétrier and Montmerle (2011); Michel et al. (2011); Montmerle (2012); Yang et al. (2022), although these studies used the strength of precipitation or the presence of fog (rather than convection) as a criterion for selection of the most appropriate covariance statistics. Here, we explore ways to determine locations of *convecting* and *non-convecting* flows. Our strategy is to determine these regions ideally before convection has fully developed, so that a data assimilation system that uses the right covariances with new observations does not lead to an analysis state that will impede the development of convection in the ensuing forecasts by imposing HB wrongly.

Seven quantities are considered here to help in the discrimination. (i) Convective available potential energy (CAPE) is a vertically integrated quantity, which can be used to help determine if an atmospheric column is 'primed' for convection. (ii)

Convective inhibition (CIN) is also a vertically integrated quantity, which can be used to help determine if a column is 're-sistant' to convection. CAPE and CIN are derived from Hydro-ABC quantities in Appendix C. (iii) The RH at any horizontal location is defined as the maximum value of $q/q_s$ in the vertical column, where $q_s$ is the saturated specific humidity defined in Sect. 2.4. Large values of RH are expected during and just before moist convection. The column maximum (iv) vertical wind, $w^{max}$, and (v) condensate, $q_c^{max}$, are important signatures of active convection, but not necessarily of pre-convective condi-

tions. (vi) hydrostatic imbalance (HI) at any horizontal location is defined as the largest absolute value of $(C\partial\tilde{\rho}'/\partial z - b')/$ $(\mathrm{RMS}(C\partial\tilde{\rho}'/\partial z) + \mathrm{RMS}(b'))$ in the column, where RMS is the root-mean-square calculated over the whole domain. A value of zero represents perfect hydrostatic balance, so convection is associated with non-zero HI. (vii) The maximum value of $\partial u/\partial x$ in a column (horizontal divergence, HD – recall there is no $y$-dependence in ABC) is the last quantity considered.

Figure 7 shows how these quantities vary with horizontal position in the domain at $t = 10\,\mathrm{min}$, which is before convection

has developed. The CAPE (red, top panel) increases towards the convection zone, but is not a smooth function of position and drops slightly at the future convection point. The CAPE increases abruptly at $\sim 0\,\mathrm{km}$ and drops at $\sim 285\,\mathrm{km}$, even though these points are within the region where $q > 0$ in the initial conditions. The CIN (blue, top) also changes abruptly at these positions, but in the opposite way: the CIN is small where the CAPE is large and vice-versa. The RH (green, top) in this case is a smooth function of position, peaking at unity around the convection zone. While the CAPE, CIN, and RH give physically reasonable

indications of where convection is possible, they do not pin-point where convection develops (namely in the convection zone).

The $w^{max}$ (red, bottom), $q_c^{max}$ (magenta, bottom), and HI (green, bottom) are fields with a finer scale than CAPE, CIN, and RH, and do peak at the precise location where convection develops. The HD (blue, bottom) is also fine-scale, but does not peak at the correct location of the future convection. The quantities $w^{max}$, $q_c^{max}$, and HI are therefore promising harbingers of convection, which may be useful for selecting vertical background error statistics that are characteristic of convection (rather

than quiescent conditions) for data assimilation. This is the case especially when the data assimilation method is otherwise non-flow-dependent, as in traditional variational schemes. Further work will be required to test the reliability of these indicators.

## 5  Conclusions

Described in this paper is how the simplified ABC model of convective-scale flow has been extended to include moist processes, namely evaporation and condensation, yielding a revised model called Hydro-ABC. This is the next step in the aim to provide

a low cost framework to study convective-scale data assimilation (DA) strategies. The following new variables, terms, and parameters have been introduced.

- The model's vapour mixing ratio field is $q$ (the gaseous phase of water), Eq. (1g).

- The model's condensate mixing ratio field is $q_c$ (the solid and/or liquid phases of water), Eq. (1h).

- The net rate of source of vapour (or equivalently the net rate of sink of condensate) is $Ev - Co$ (the evaporation minus

the condensation rates) for each grid point. Only the difference needs to be known and not the values of $Ev$ and $Co$ separately.

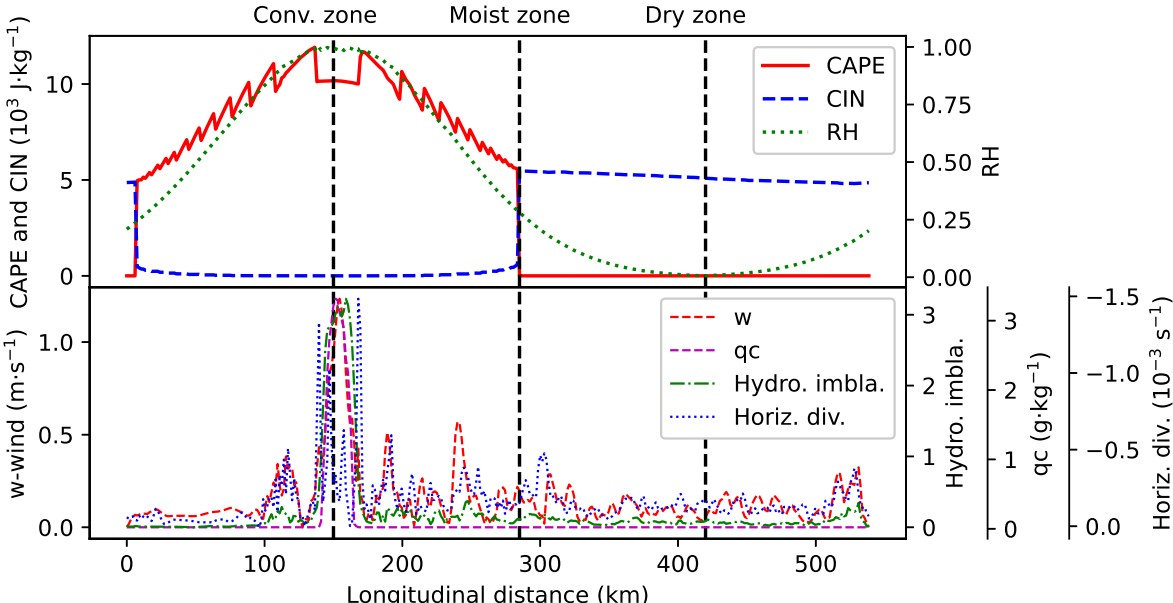

**Figure 7.** The geographical variation of various indicators/harbingers of convection at $t = 10\,\mathrm{min}$. Plotted in the top panel are: convective available potential energy (red line), convective inhibition (blue), and relative humidity (green). Plotted in the bottom panel are column maximum values of: vertical wind (red), condensate mixing ratio (magenta), hydrostatic imbalance (green), and horizontal divergence (blue). The black vertical lines mark locations that are within the convection, moist, and dry zones (see Fig. 2).

- Latent heat is released (absorbed) for net condensation (evaporation). The latent heating rate is $\mathcal{S}_{b'}$, and is a source term for buoyancy (Eq. (1e)) (latent heat is assumed to be exchanged directly only with buoyancy energy). $\mathcal{S}_{b'}$ and $Ev - Co$ are not determined explicitly, but are the result of potentially solving a non-linear system of equations as set out in Sect. 2.3. The latent heat of vapourisation, $L_v$ (set to $2500\,\mathrm{Jg}^{-1}$) is introduced which relates changes of vapour to changes in latent heat energy, Eq. (3).

- Condensation happens when situations are super-saturated ($q > q_s$), in which case $q$ drops immediately (practically over a model timestep) to a new $q_s$, and $b'$ is increased from its previous value due to latent heat release. This is the reason for the non-linear equations as mentioned above, see Eq. (7). When $b' < 0$ condensation is not allowed when the ratio of latent heat release to buoyancy energy is greater than $\Gamma$ (set to 10). This is to avoid excessively large jumps to buoyancy.

- Evaporation happens when situations are sub-saturated, when there is a finite amount of condensate to evaporate, and when $b'$ is positive. The last condition is imposed as otherwise $b'$ will increase (i.e. become less negative) when evaporation happens, which is unphysical. Evaporation happens at a slower rate (timescale $\tau$, set to $1000\,\mathrm{s}$) than condensation, see Eq. (6).

- The saturated mixing ratio, $q_s$, sets a maximum $q$ that can be supported. In real systems $q_s$ is a function of temperature and pressure. Since these quantities do not exist in the ABC model, analogous quantities are derived based on an assumed reference density profile, hydrostatic balance, and the correspondence between buoyancy and potential temperature. A formula is derived for $q_s$ in terms of ABC variables $b'$ and $\tilde{\rho}'$ as set out in Sect. 2.4. The conditions used to relate $b'$ and $\tilde{\rho}'$ to $q_s$ do not need to be exactly relevant to the ABC system, but are used only as a basis for a sensible relationship.

Hydro-ABC is constructed to satisfy certain conservation properties over the whole domain. These are for total water ($q + q_c$, Sect. 2.2.1) and total moist energy (defined as dry energy plus latent heat energy, Sect. 2.2.2). There are no precipitation processes in Hydro-ABC; this is to keep the system as simple as possible while allowing highly non-linear processes.

An example run of Hydro-ABC shows a strong convective event driven by latent heat release. This forms a realistic anvil cloud (Sect. 3.1), and the excitation of inertio-gravity and acoustic modes, higher in frequency than those normally generated by the dry ABC model (Sect. 3.3).

A very important concern in DA applied to convective-scale flows involves how the background error covariance statistics are affected by the flow situation. This is an issue in real forecasting and DA systems where attempts have been made in the literature to use statistics that depend on the presence of background properties like rain and fog. This study gives an example of this by demonstrating how flow-dependent statistics (found from a 93-member ensemble) can depend on the presence of convection. It is found that there can be significant differences between the covariances in convecting and non-convecting regions, including the breakdown of hydrostatic balance. Many operational systems base their static background error covariance model, or the training data used to calibrate it, on hydrostatic balance (Berre, 2000; Bannister, 2008; Gustafsson et al., 2018; Bannister et al., 2020), and so this result may urge changes to this practice.

Given that the covariances depend on convection, it is necessary to be able to diagnose reliably the presence of convection (or impending convection) especially if a suitable ensemble is not available, and one needs to instead prescribe covariances for the DA. Such an indicator can be applied to the background state in a DA application, in order to decide the most relevant set of error covariances. Using climatologically-derived (static) error covariances, or covariances that support hydrostatic flow, may well impede the progress of convection prediction. A number of convection diagnostics have been proposed here as an indicator or harbinger of convection, namely relative humidity, hydrostatic imbalance, horizontal divergence, convective available potential energy, convective inhibition, vertical wind, and condensate mixing ratio (Sect. 4.2). It is found in the example shown that all of these quantities are sensitive to convection, but hydrostatic imbalance, vertical wind, and condensate mixing ratio seem to be the best indicator of pin-pointing impending convection. Further work though is required to decide which quantity or combination of quantities is best, and which threshold values should be used. This will be part of the next stage of this work, namely to adapt the dry ABC-DA system to assimilate water-related information. Another interesting avenue that may be explored is to add a precipitation process (i.e. an additional fast downward transport of condensate), which could give rise to cold pools.

## Appendix A: Continuous total energy equations of Hydro-ABC

The formulation of Hydro-ABC is based on a time-step split into a dynamical core and a micro-physics parametrisation. Hydro-ABC can also be represented as continuous equations, i.e. as Eqs. (1), but with (1e) replaced with

$$\frac{\partial b'}{\partial t} + B\boldsymbol{u} \cdot \nabla b' + A^2 w = -\frac{A^2 L_v}{b'}(Ev - Co), \tag{A1}$$

which is an explicit form of the buoyancy equation. This is consistent with the basis of our micro-physics parametrisation, Eq. (5), where $\Delta^{\mathrm{mp}}\left(b'^2\right) = \frac{\partial}{\partial t}\left(b'^2\right)\Delta t = 2b'\frac{\partial}{\partial t}\left(b'\right)\Delta t = 2b'\Delta^{\mathrm{mp}}b'$. Plugging this into (5) gives

$$\Delta^{\mathrm{mp}}b' = -\frac{A^2 L_v}{b'}(Ev - Co)\Delta t,$$

i.e. the right-hand-side of (A1). In order to show that this leads to an energy conserving system of equations, multiply (A1) by $\tilde{\rho}b'/A^2$:

$$\frac{\tilde{\rho}b'}{A^2}\frac{\partial b'}{\partial t} + B\frac{\tilde{\rho}b'}{A^2}\boldsymbol{u} \cdot \nabla b' + \tilde{\rho}b'w \;\; = \;\; -\tilde{\rho}L_v(Ev - Co)$$

$$\tilde{\rho}\frac{\partial\left[b'^2/(2A^2)\right]}{\partial t} + B\tilde{\rho}\boldsymbol{u} \cdot \nabla\left[b'^2/(2A^2)\right] + \tilde{\rho}b'w \;\; = \;\; -\tilde{\rho}L_v(Ev - Co).$$

Then use the following identity (Petrie et al. (2017), their Eq. (20)):

$$\tilde{\rho}\left(\frac{\partial \gamma}{\partial t} + B\boldsymbol{u} \cdot \nabla\gamma\right) = \frac{\partial(\tilde{\rho}\gamma)}{\partial t} + B\nabla \cdot (\tilde{\rho}\gamma\boldsymbol{u}), \tag{A2}$$

with $\gamma = b'^2/(2A^2)$:

$$\frac{\partial\left[\tilde{\rho}b'^2/(2A^2)\right]}{\partial t} + B\nabla \cdot \left[\tilde{\rho}b'^2/(2A^2)\boldsymbol{u}\right] + \tilde{\rho}b'w = -\tilde{\rho}L_v(Ev - Co).$$

Making the substitution $E_b = \tilde{\rho}b'^2/(2A^2)$ (see Sect. 2.2.2), gives:

$$\frac{\partial E_b}{\partial t} + B\nabla \cdot [E_b\boldsymbol{u}] + \tilde{\rho}b'w = -\tilde{\rho}L_v(Ev - Co). \tag{A3}$$

This is the continuous form of the equation for buoyancy energy. Next multiply (1g) by $L_v$:

$$\frac{\partial \tilde{\rho}L_v q}{\partial t} + B\nabla \cdot (\tilde{\rho}L_v q\boldsymbol{u}) = \tilde{\rho}L_v(Ev - Co), \tag{A5}$$

and make the substitution (3):

$$\frac{\partial E_l}{\partial t} + B\nabla \cdot (E_l\boldsymbol{u}) = \tilde{\rho}L_v(Ev - Co). \tag{A4}$$

This is the continuous form of the equation for latent heat energy. The equations for kinetic and elastic energies ($E_k$ and $E_e$ respectively) are derived in a similar way to (A3) (see Petrie et al. (2017), their Sect. 2.3):

$$\frac{\partial E_k}{\partial t} + B\nabla \cdot (E_k\boldsymbol{u}) - \tilde{\rho}b'w + C\tilde{\rho}\boldsymbol{u} \cdot \nabla\tilde{\rho}' \;\; = \;\; 0, \tag{A5}$$

$$\frac{\partial E_e}{\partial t} + C\tilde{\rho}'\nabla \cdot (\tilde{\rho}\boldsymbol{u}) \;\; = \;\; 0. \tag{A6}$$

Adding (A3), (A4), (A5), and (A6) leads to the equation for the total moist energy, $E_{\mathrm{moist}} = E_{\mathrm{k}} + E_{\mathrm{b}} + E_{\mathrm{e}} + E_{\mathrm{l}}$:

$$\frac{\partial E_{\mathrm{moist}}}{\partial t} + B\nabla \cdot \left[(E_{\mathrm{b}} + E_{\mathrm{k}} + E_{\mathrm{l}})\,\boldsymbol{u}\right] + C\nabla \cdot (\tilde{\rho}'\tilde{\rho}\boldsymbol{u}) = 0.$$

Integrating this quantity over the whole domain, using the divergence theorem, and exploiting the horizontal periodic boundary conditions, and the zero vertical wind at the top and bottom boundaries (see Petrie et al. (2017), their Sect. 3.2), the following is found

$$\frac{\partial}{\partial t}\left(\int\int dx dz\, E_{\mathrm{moist}}\right) = 0, \tag{A7}$$

proving that the integral of total moist energy is conserved under the governing equations (1) and (A1).

The issue with Eq. (A1) is the $1/b'$ factor on the right-hand-side. This issue is avoided by using the micro-physics parametrisation step represented by Eq. (5).

### Appendix B: Pseudo code for the micro-physics parametrisation

Annotated pseudo code describing the dynamics/micro-physics parametrisation is given in this appendix.

1. **Propagate** the Hydro-ABC fields $u$, $v$, $w$, $\tilde{\rho}'$, $b'$, $q$, and $q_{\mathrm{c}}$ (described by Eqs. (1)) by one timestep.

2. **Calculate** the pre-microphysics saturated specific humidity field, $q_{\mathrm{s}}(x, z)$, Eq. (9). This requires the corresponding pressure, $p(x, z)$, Eq. (10) and temperature, $T(x, z)$, Eq. (12), which are related to model variables. How the micro-physics behaves depends on Eq. (6).

3. **If** $q(x, z) > q_{\mathrm{s}}(x, z)$ condensation may happen (see points 1a and 1b in Sect. 2.3). The following are performed for each $x, z$.

   (a) **If** $b' \geq 0$ allow condensation to happen by solving the following non-linear simultaneous equations:
   $$\begin{aligned} b'_{\mathrm{mp}} &= +\sqrt{b'^2 + 2A^2 L_{\mathrm{v}}\,(q - q_{\mathrm{mp}})} \\ q_{\mathrm{mp}} &= q_{\mathrm{s}}(b'_{\mathrm{mp}}), \end{aligned}$$
   and then setting $q_{\mathrm{cmp}} = q_{\mathrm{c}} + (q - q_{\mathrm{mp}})$.

   (b) **Else if** $\tilde{\rho} L_{\mathrm{v}}\,|q - q_{\mathrm{s}}|/E_{\mathrm{b}} > \Gamma$, then the latent heat energy released by condensation will be much greater than the current buoyancy energy. This will allow condensation to happen by setting:
   $$\begin{aligned} b'_{\mathrm{mp}} &= +\sqrt{b'^2 + 2A^2 L_{\mathrm{v}}\,(q - q_{\mathrm{s}})} \\ q_{\mathrm{mp}} &= q_{\mathrm{s}} \\ q_{\mathrm{cmp}} &= q_{\mathrm{c}} + (q - q_{\mathrm{mp}}). \end{aligned}$$

   (c) **Else** do nothing (to avoid unrealistic situations).

4. **Else if** $b' > 0$ evaporation may happen (see point 2 in Sect. 2.3 and Eq. (6)). The following are performed for each $x, z$. Calculate three positive semi-definite quantities $((q_s - q)(1 - e^{-\Delta t/\tau}))$, $q_c$, and $b'^2/(2A^2 L_v))$ and find which has the smallest value.

    (a) **If** $(q_s - q)(1 - e^{-\Delta t/\tau})$ is the smallest then there is enough condensate and buoyancy energy in the grid box to allow this amount of evaporation to occur by setting:

$$\begin{aligned}
b'_{\text{mp}} &= +\sqrt{b'^2 - 2A^2 L_v (q_s - q)(1 - e^{-\Delta t/\tau})} \\
q_{\text{cmp}} &= q_c - (q_s - q)(1 - e^{-\Delta t/\tau}) \\
q_{\text{mp}} &= q + (q_s - q)(1 - e^{-\Delta t/\tau}).
\end{aligned}$$

    (b) **If** $q_c$ is the smallest then evaporation is limited only by availablility of condensate (there is enough buoyancy energy available to allow all of this condensate in the grid box to evaporate) by transferring all condensate to vapour, leaving:

$$\begin{aligned}
b'_{\text{mp}} &= +\sqrt{b'^2 - 2A^2 L_v q_c} \\
q_{\text{cmp}} &= 0 \\
q_{\text{mp}} &= q + q_c.
\end{aligned}$$

    (c) **Else** $b'^2/(2A^2 L_v)$ is the smallest then evaporation is limited only by availability of buoyancy energy, $E_b = b'^2/(2A^2\tilde{\rho})$ (there is enough condensate available) by transferring all buoyancy energy to latent heat, leaving:

$$\begin{aligned}
b'_{\text{mp}} &= 0 \\
q_{\text{cmp}} &= q_c - b'^2/(2A^2 L_v) \\
q_{\text{mp}} &= q + b'^2/(2A^2 L_v).
\end{aligned}$$

5. **Overwrite** the model's variables, $b'$, $q$, and $q_c$ with their post micro-physics values, $b'_{\text{mp}}$, $q_{\text{mp}}$, and $q_{\text{cmc}}$ respectively.

## Appendix C: Computation of the convective available potential energy (CAPE)

In the real atmosphere CAPE is proportional to the vertical integral of the difference between a moist parcel's temperature, $\hat{T}$, and the environmental temperature, $T_{\text{env}}$ (e.g. Sect. 7.4.1 of Salby (1996), $\text{CAPE} = g \int_{\text{LFC}}^{\text{LNB}} (\hat{T} - T_{\text{env}})/T_{\text{env}} \, dz$, where LFC is the level of free convection and LNB is the level of neutral buoyancy – see below). CAPE represents the amount of energy that can be converted to kinetic energy by convective instability. A positive value indicates energy is available for convection, and is thus a measure of instability. The above equation for CAPE is relevant to the standard equations of motion. In Hydro-ABC there is no explicit temperature variable, but we can derive a physically reasonable CAPE-like quantity by analysing the energetics of an air parcel using the ABC equations.

Consider an air parcel, whose properties are indicated with a 'hat'. The Lagrangian derivative of $\hat{w}$ is

$$\frac{\partial \hat{w}}{\partial t} + B\hat{\mathbf{u}} \cdot \nabla \hat{w} = \frac{D\hat{w}}{Dt} = \frac{D^2 \hat{z}}{Dt^2}, \tag{C1}$$

where $\hat{z}$ is the vertical position of the parcel. Equation (C1) is the vertical acceleration of the parcel. We can substitute for $\partial \hat{w}/\partial t + B\hat{\mathbf{u}} \cdot \nabla \hat{w}$ using the vertical momentum equation (1c). The parcel equation is then

$$\frac{D^2 \hat{z}}{Dt^2} = -C\frac{\partial \hat{\tilde{\rho}}}{\partial z} + \hat{b}'. \tag{C2}$$

The environmental profile (the mean state around the convecting column, found by averaging the model fields over 101 points horizontally centred on the profile of interest) is found to be in hydrostatic equilibrium (not shown):

$$-C\frac{\partial \tilde{\rho}'_{\mathrm{env}}}{\partial z} + b'_{\mathrm{env}} = 0. \tag{C3}$$

Taking the difference between (C2) and (C3) gives

$$\begin{aligned}
\frac{D^2 \hat{z}}{Dt^2} &= -C\frac{\partial \left(\hat{\tilde{\rho}} - \tilde{\rho}'_{\mathrm{env}}\right)}{\partial z} + \left(\hat{b}' - b'_{\mathrm{env}}\right) \\
&\approx \hat{b}' - b'_{\mathrm{env}},
\end{aligned} \tag{C4}$$

where in the second line we have assumed mechanical equilibrium (that $\hat{\tilde{\rho}} = \tilde{\rho}'_{\mathrm{env}}$). This gives a formula for the parcel's acceleration (or specific force), which is used in the CAPE formula below.

The parcel's buoyancy profile, $\hat{b}'(z_i)$, is found by raising a hypothetical parcel from the surface. At the surface the parcel's buoyancy and vapour mixing ratio take the model's values, $\hat{b}'(z_0) = b'(z_0)$ and $\hat{q}(z_0) = q(z_0)$ respectively. Start with level $i = 1$. By raising the parcel to the next level, the total buoyancy is assumed to be conserved, $\hat{b}(z_i) = \hat{b}(z_{i-1})$. The total buoyancy is the sum of the reference state and the perturbation, namely $\hat{b}(z_i) = \hat{b}_0(z_i) + \hat{b}'(z_i)$. By definition (Petrie et al., 2017), $A^2$ is the rate of increase of reference buoyancy with height, namely $\hat{b}_0(z_{i+1}) - \hat{b}_0(z_i) = A^2(z_{i+1} - z_i)$ in discrete form. Putting the last three equations together gives the parcel's perturbation value at $z_i$:

$$\hat{b}'(z_i) = \hat{b}'(z_{i-1}) - A^2(z_i - z_{i-1}). \tag{C5}$$

The parcel's mixing ratio is also assumed to be conserved, $\hat{q}(z_i) = \hat{q}(z_{i-1})$.

Should the parcel become supersaturated, the same non-linear equations as used by hydro ABC's micro-physics scheme to adjust $\hat{b}'(z_i)$ and $\hat{q}(z_i)$ towards saturation are used here. This calculates revised values of $\hat{b}'(z_i)$ and $\hat{q}(z_i)$ (call these $\hat{b}'_{\mathrm{mp}}(z_i)$ and $\hat{q}_{\mathrm{mp}}(z_i)$ respectively). The equations are given in point 1a of Sect. 2.3, where a 'hat' should be added to those equations to make them relevant here. Once $\hat{b}'_{\mathrm{mp}}(z_i)$ and $\hat{q}_{\mathrm{mp}}(z_i)$ have been found, $\hat{b}'(z_i)$ and $\hat{q}(z_i)$ are then overwritten respectively by these adjusted values. This process is repeated by replacing $z_i \to z_{i+1}$ until the top of the model is reached.

Once this procedure is complete, two special levels are found. Starting from the bottom of the domain, the first level encountered that has the property $\hat{b}' \geq b'_{\mathrm{env}}$ is called the level of free convection, $z_{\mathrm{LFC}}$, and the first level above that where $\hat{b}' \leq b'_{\mathrm{env}}$

is called the level of neutral buoyancy, $z_{\mathrm{LNB}}$. The CAPE is then computed from the integral

$$\mathrm{CAPE} = \int_{z_{\mathrm{LFC}}}^{z_{\mathrm{LNB}}} \left( \hat{b}'(z) - b'_{\mathrm{env}}(z) \right) dz, \tag{C6}$$

which has units of $\mathrm{Jkg}^{-1}$. The related quantity convective inhibition (CIN) can also be found from the parcel's profile:

$$\mathrm{CIN} = - \int_{0}^{z_{\mathrm{LFC}}} \left( \hat{b}'(z) - b'_{\mathrm{env}}(z) \right) dz. \tag{C7}$$

A large CAPE is associated with large positive differences $\hat{b}'(z) - b'_{\mathrm{env}}(z)$ between $z_{\mathrm{LFC}}$ and $z_{\mathrm{LNB}}$, which is associated with buoyant air (air locally warmer than its surroundings). Similarly a large value of CIN is associated with the reverse – cooler air at a given location compared to its surroundings.

Should a value of $z_{\mathrm{LFC}}$ not be found (owing to $\hat{b}' < b'_{\mathrm{env}}$ for all levels), CAPE would be zero, and CIN would be found by integrating over the entire profile. Should a value of $z_{\mathrm{LNB}}$ not be found (owing to $\hat{b}' > b'_{\mathrm{env}}$ for all levels above $z_{\mathrm{LFC}}$), then CAPE would be found by integrating to the top of the model.

Example profiles of the parcel and environmental buoyancy are shown in Fig. C1 for three positions in the domain at $t = 10\,\mathrm{min}$. In the convection zone at $148.5\,\mathrm{km}$ (left panel), $\hat{b}' > b'_{\mathrm{env}}$ for all levels, so the CAPE has a large value, consistent with the strong convection there. Two further profiles are shown for the moist zone. One, at $271.5\,\mathrm{km}$ (middle panel), is where the CAPE remains large but where a value of $z_{\mathrm{LFC}}$ is found. There is therefore a non-zero CIN associated with this position, but the moist instability still has influence. The other, at $286.5\,\mathrm{km}$ (right panel), is just past the position where the CAPE drops away (see Fig. 7). Here $\hat{b}' < b'_{\mathrm{env}}$ for all levels, so there is no CAPE, but a large value of CIN. This is the situation into the dry zone too, where a very similar plot is found, for instance at $420\,\mathrm{km}$ (not shown).

*Code availability.* The Hydro-ABC V2.0 code is written in Fortran-90 and is freely available via a GitHub repository (Zhu and RN, 2022). The Hydro-ABC V2.0 user documentation is available in the same place.

*Author contributions.* JZ and RNB jointly designed the moisture framework. JZ modified the dry ABC code to include the moist processes described in this paper, ran the model and produced Figs. 2, 3, 4, 5, 6, 7, and C1. RNB wrote the paper and produced Fig. 1.

*Competing interests.* The authors declare that they have no competing interests.

*Acknowledgements.* The authors would like to thank the National Natural Science Foundation of China for funding JZ (grant ref 41875136), and the Natural Environment Research Council, which provides national capability funding for RNB via the National Centre for Earth

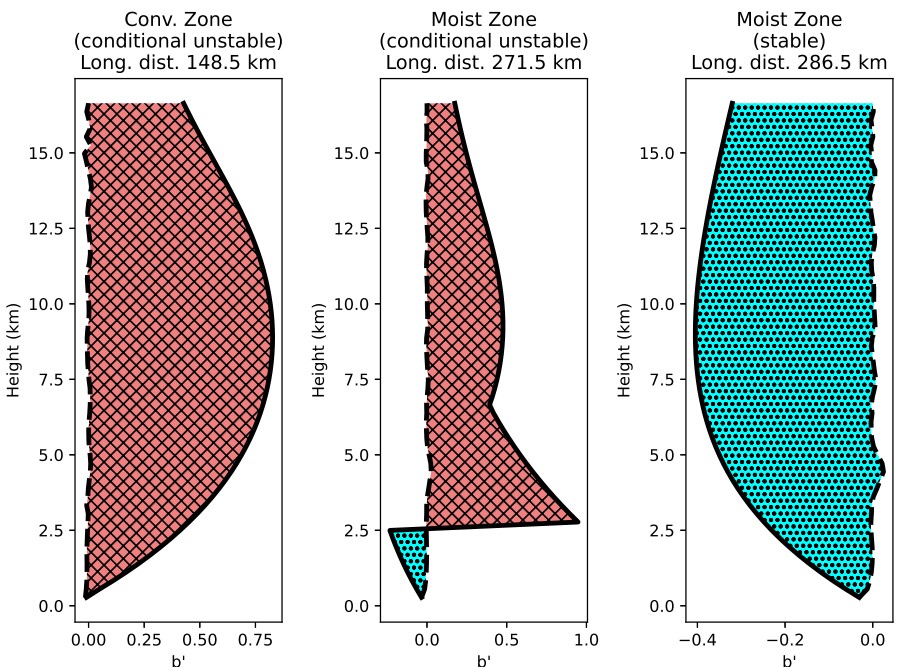

**Figure C1.** Example profiles of a parcel's computed profile ($\hat{b}'$, solid line) and the environmental profile ($b'_{\text{env}}$, dashed line) for three horizontal positions in the domain (calculated at $t = 10\,\text{min}$). The horizontal positions are in the convection zone ($x = 148.5\,\text{km}$), and two points within the moist zone ($271.5$, and $286.5\,\text{km}$). The area of the red hatched region is the CAPE, and the area of the turquoise hatched region is the CIN.

Observation (contract number PR140015). The authors would also like to thank Mike Cullen for correspondence during the development of
630  Hydro-ABC, and two anonymous reviewers for providing valuable and insightful feedback with their reviews.

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
