# Peer review of "The "Hydro-ABC model" (Vn 2.0): a simplified convective-scale model with moist dynamics"

_EGUsphere, 2022_

## Author Comment (AC1)

**Responses to Reviewer 1**

We are very grateful to reviewer 1 for his/her/their comments on our manuscript. We have considered the comments carefully, and we hope that the reviewer and editor are happy with our responses. The reviewer's report is reproduced below (in red), together with our responses. Where changes have already been made to the manuscript, these are underlined.

**Comments of Reviewer 1**

This manuscript extends the existing dry ABC model to include mixing ratios of water vapor and cloud water, termed the 'hydro-ABC model'. This manuscript introduces and evaluates the extended equations for the 'hydro-ABC model'. The success of simulation for a realistic-looking anvil cloud offers the possibility of exploring convective-scale DA strategies within a low-cost model. Investigations of error statistics are also conducted. Generally, this manuscript is a complete work and provides a cheaper approach to studying convective-scale DA. My overall recommendation is for acceptance pending minor revisions.

- Many thanks to the reviewer for the very positive and insightful comments. We have numbered the reviewer's points below, with our responses.

1. Line 7: "-"is missing after "Hydro". Change made.

2. Line 101: Check the equation. The first "=" sign seems to be "+"? Change made.

3. Fig. 5: I am confused about the vertical correlations of w with q and qc at 80 and onward. Both correlations show strong positive values in the low levels. Intuitively, the increased qc means the reduced q. Why do we see positive correlations for both of them? This is a very interesting point. We have added the following text (underlined): "This suggests that the region of positive $q_c$-$w$ covariances just mentioned is not caused by advection (the covariances would have negative sign if they were), but instead by the process of more condensate being produced in association with stronger $w$. Note that in most parts of the cloud layer $q$ and $q_c$ are both positively correlated with $w$ (and in fact positively correlated with each other, not shown), which may appear counter-intuitive. In the convection zone, convergence and condensation processes are happening. The convergence can increase $q$ while the condensation will decrease $q$, but increase $q_c$. As long as the increase of $q$ caused by convergence is larger than the loss of $q$ by condensation, there is a net increase in $q$, in which case both $q$ and $q_c$ could have a positive correlation with the vertical wind."

4. The proposed "hydro-ABC model" does not include the precipitation processes. However, the precipitation can lead to the development of cold pools, which facilitate the maintenance of convections. Will the authors have plans to include the processes related to cold pools in the proposed model? We have added the following text to the end of the paper: "Another interesting avenue that may be explored is to add a precipitation process (i.e. an additional fast downward transport of condensate), which could give rise to cold pools."

5. Figure 6: It is unclear how the authors define the three zones in this figure. Because Figure 7 suggests that the authors attempt to find indicators for determining the three zones. Three zones are defined ("convection", "moist", and "dry") in order to choose representative points for study in Fig. 7.

    (a) Labels of the zones have been added to Fig. 2, and the following text has been added to the caption, "Three zones are defined, the convection zone (columns close to the convection), the moist zone (columns of non-zero moisture in the initial conditions), and the dry zone (zero moisture in the initial conditions)."

    (b) The following text has been added to the discussion of Fig. 6 (underlined), "Figure 6 shows vertical correlations between a different selection of quantities. The top three rows show auto-correlations of $\tilde{\rho}'$ for the convection, moist, and dry zones respectively (see Fig. 2 for definition of the zones where the columns are selected, and Fig. 6 caption for their specific lateral positions)."

[Figure]

Figure 7: The geographical variation of various indicators/harbingers of convection at $t = 10$min. Plotted in the top panel are: convective available potential energy (red line), convective inhibition (blue), and relative humidity (green). Plotted in the bottom panel are column maximum values of: vertical wind (red), condensate mixing ratio (magenta), hydrostatic imbalance (green), and horizontal divergence (blue). The black vertical lines mark locations that are within the convection, moist, and dry zones (see Fig. 2).

      (c) The following text has been changed in Fig. 7, "The black vertical lines mark locations that are within the convection, moist, and dry zones (see Fig. 2)."

6. Figure 7: I suggest using qc, w, or their variants, and they may be good potential choices for the indicators as well. Figure 7 has been modified to include these quantities (see Fig.), and discussion has been added to the revised paper as follows:

To recap, CAPE is "Convective Available Potential Energy", CIN is "Convective INhibition", RH is "Relative Humidity", HI is "Hydrostatic Imbalance", and HD is "Horizontal Divergence".

"Figure 7 shows how these quantities vary with horizontal position in the domain at $t = 10$min, which is before convection has developed. The CAPE (red, top panel) increases towards the convection zone, but is not a smooth function of position and drops slightly at the future convection point. The CAPE increases abruptly at $\sim$ 0km and drops at $\sim$ 285km, even though these points are within the region where $q > 0$ in the initial conditions. The CIN (blue, top) also changes abruptly at these positions, but in the opposite way: the CIN is small where the CAPE is large and vice-versa. The RH (green, top) in this case is a smooth function of position, peaking at unity around the convection zone. While the CAPE, CIN, and RH give physically reasonable indications of where convection is possible, they do not pin-point where convection develops (namely in the convection zone).

The $w^{\mathrm{max}}$ (red, bottom), $q_{\mathrm{c}}^{\mathrm{max}}$ (magenta, bottom), and HI (green, bottom) are fields with a finer scale than CAPE, CIN, and RH, and do peak at the precise location where convection develops. The HD (blue, bottom) is also fine-scale, but does not peak at the correct location of the future convection. The quantities $w^{\mathrm{max}}$, $q_{\mathrm{c}}^{\mathrm{max}}$, and HI are therefore promising harbingers of convection, which may be useful for selecting vertical background error statistics that are characteristic of convection (rather than quiescent conditions) for data assimilation. This is the case especially when the data assimilation method is otherwise non-flow-dependent, as in traditional variational schemes. Further work will be required to test the reliability of these indicators."

---

## Author Comment (AC2)

**Responses to Reviewer 2**

We are very grateful to reviewer 2 for his/her/their very thorough reading of our manuscript. We have considered the comments carefully, and we hope that the reviewer and editor are happy with our responses. The reviewer has provided two reports. The first is the outline and is reproduced in part 1 below (in red), together with our responses. The second is more detailed in the form of an annotated PDF and each comment from there has been reproduced in part 2 below (again in red), together with our responses. Where changes have already been made to the manuscript, these are underlined.

**1 Main comments of Reviewer 2**

The manuscript of Zhu and Bannister presents a novel, moist-dynamics version of the ABC-model, the new model is called Hydro-ABC, with the aim to perform future DA simulations for testing (new) DA schemes. The Hydro-ABC is a reduced model (no y-derivatives) relative to NWP models and it supports vortical dynamics, gravity and acoustic waves, as well as simplified moist dynamics. The moist dynamics included concerns the dynamics of water vapour and condensate fields and the model is constructed in such a way that the total energy including the moist component is conserved. The authors present preliminary (ensemble) simulations including correlations between various fields of interest. These developments for DA simulations are very interesting and such a reduced moist dynamics approach deserves due attention. So this is excellent news.

- Many thanks to the reviewer for the very positive comments.

Yet the work described to date in the manuscript has some shortcomings. The manuscript requires therefore seemingly some modifications, with the following suggestions for improvement:
    The formulation of the moist dynamics (in section 2.3) is rather implicit and unclear, and immediately leads to a parameterisation that seems to hinge on the discrete time-step splitting between dry and moist dynamics. The dynamics should be described first on a continuum level before mixing it up with the time-step splitting. Whether the description is implicit or not, e.g. for S_b', in the end, a closed-form formulation should be reached/available, otherwise a numerical implementation would not be achieved or possible. (1) What is required is a complete, final closed-form description of the entire continuum dynamics, presumably including implicit relations to be solved. Presently, section 2.3 is rather unclear and it is difficult if not impossible to figure out what the final formulation is. E.g., the implied relation for S_b' seems to become 0/0 when b'=0, which presumably can be fixed via some l'Hopital rule being invoked implicitly?

- Many thanks to the reviewer for the constructive and insightful comments.

- A new appendix (new Appendix A) provides the continuous equations for Hydro-ABC, and shows that the total moist energy is conserved. This will be available in the revision of the paper.

- The scheme that has emerged from our study did indeed turn out to be more complicated that we had anticipated, owing partly to a singularity in $b'$ in the underlying continuum equations (linked to the issue raised above). This is the reason we decided to instead pursue a dynamical core + micro-physics time-stepping (generally in line with operational models), which we have shown does work in practice.

- We have modified the text, which we hope will make the micro-physics scheme presentation clearer (see detailed points 8-17 in part 2 below).

- We don't agree though that a closed form formulation of $\mathcal{S}_{b'}$ needs to be available to be valid. Like many non-linear equations, an analytical solution is not available here, and so the solution requires implicit numerical treatment.

- The issue with $\mathcal{S}_{b'}$ becoming 0/0 when $b' \to 0$ comes from analysis of Eq. (5) in the paper:

$$\Delta^{\mathrm{mp}}\left(b'^{2}\right) = -2A^{2}L_{\mathrm{v}}\left(Ev - Co\right)\Delta t$$
$$\Rightarrow b'\Delta^{\mathrm{mp}}\left(b'\right) \approx -A^{2}L_{\mathrm{v}}\left(Ev - Co\right)\Delta t$$
$$\text{so } \Delta^{\mathrm{mp}}\left(b'\right) \approx -A^{2}L_{\mathrm{v}}\left(Ev - Co\right)\Delta t/b',$$

(note $\mathcal{S}_{b'}$ is defined as the rate of change of $b'$ due to micro-physics, $\Delta^{\mathrm{mp}}\left(b'\right)/\Delta t$, so the above leads to $\mathcal{S}_{b'} \approx -A^2 L_{\mathrm{v}}\left(Ev - Co\right)/b'$). This result is applicable to infinitesimal changes, where $\Delta^{\mathrm{mp}}\left(b'^2\right) \approx \left[\frac{\partial}{\partial t}\big|_{\mathrm{mp}}\left(b'^2\right)\right]\Delta t = 2b'\left.\frac{\partial b'}{\partial t}\right|_{\mathrm{mp}}\Delta t = 2b'\Delta^{\mathrm{mp}}\left(b'\right)$. The problem of dividing by zero can be avoided, however, by taking finite changes, which leads to Eq. (7).

(2) In addition, it would be useful to have a (pseudo)code available of the moist-dynamics algorithm, which would presumably involve the split time-stepping. I cannot follow the second option in (6) at all. The time-stepping scheme used and the time-splitting should be defined and as such explained better.

- Thank you for your advice. We have added the pseudo code as an appendix (new Appendix B). With our other modifications to the paper, we hope this makes the scheme clearer. This will be available in the revision of the paper.

(1) There is an accumulation of minor issues, including a few sentences and statements which are nearly or completely incomprehensible. An anonymised, annotated manuscript is made available to assess and address these.

- Thank you for the details comments. These have been very useful.

- These are addressed in the detailed comments in part 2 below.

(2) Also note the various queries and remarks therein on resolution and timestep issues. Please consider better explaining figure 6 in the text; please tie statements in the text with a clear pointing out in how the reader can discern the statement from an identified observational feature in the figure. At the moment it is difficult to link text to features in the figure.

- Based on the detailed comments in the annotated manuscript, we have made changes, in particular to the part concerning signatures of hydrostatic balance. See point 29 in part 2.

CIN does not seem to reveal very much in contrast to what is stated.

- See point 31 in part 2.

In Appendix A, CAPE and CIN for hydro-ABC are defined, after noting that there is no T and T_environmental available in the model; yet in eqn (12) a temperature is defined, used earlier in q_s, which could serve as such? Why is the approach followed to define CAPE and CIN valid and how does this link to an approach using the T of eqn (12) and its T_environmental analog. Why is the algorithm to define "environmental variable" valid; can one prove that (numerically); how does it depend on the chosen averaging? While the goals and approaches followed in the presented research are great, and the simulations promising, the above queries seem to warrant some revisions to address the above queries.

- Note that the original Appendix A is Appendix C in the revision.

- The reason for using the form of the CAPE presented instead of using the standard version of CAPE is discussed in point 40 in part 2.

- The technique of using averaging to estimate an environmental profile is standard. We do not seek to prove this is an accurate approach, just to show empirically that it is a potentially useful indicator of convection. See points 38 and 39 in part 2.

Kind regards.

**2 Specific comments of reviewer 2 (from the annotated version of the paper in the supplement)**

The following are based on comments in the reviewer's supplement (annotated PDF). Where small changes have been added to the text it is underlined.

1. Abstract, "singular given "conditions"?" "hinders nor hastens" is changed to "hinder nor hasten".

2. Abstract, "latter part of this sentence should be logically linked to the next sentence; reword please, i.e. the simplified model meant is the ABC-model" and "also twice a sentence starting with "this" rephrase.": 'Solving this problem is difficult and expensive using operational-scale numerical weather prediction systems, and so a simplified model of convective-scale flow is under development. This paper extends the "ABC model" of dry convective-scale flow to include mixing ratios of vapour and condensate phases of water. The revised model is called "Hydro-ABC".'
changed to
'Solving this problem is difficult and expensive using operational-scale numerical weather prediction systems, and so a simplified model of convective-scale flow is under development (called the "ABC model"). This paper extends the existing ABC model of dry convective-scale flow to include mixing ratios of vapour and condensate phases of water. The revised model is called "Hydro-ABC".'

3. Abstract, "if the "next stage" is beyond the present paper it is odd to mention that in an abstract; if the "next stage" is dealt with herein that should be made clear; rephrasing needed.": "This behaviour means that Hydro-ABC is a challenging model to allow experimentation with innovative data assimilation strategies in the next stage of work. Further, an ensemble of Hydro-ABC integrations is also performed in order to study the possible forecast error covariance statistics (necessary for data assimilation)."
changed to
"This behaviour means that Hydro-ABC is a sufficiently challenging model which will allow experimentation with innovative data assimilation strategies in future work. An ensemble of Hydro-ABC integrations is performed in order to study the possible forecast error covariance statistics (knowledge of which is necessary for data assimilation)."

4. Abstract: "is" changed to "will be".

5. Near end of Sect. 1, sentence has been changed (including correction of typo indicated by the reviewer): "Until now however, the ABC model lacks a water variable, meaning that studies have been limited to dry dynamics only, where the behaviour of the model shows little evidence of sporadic convective behaviour."
changed to
"Until now however, the ABC model lacks a water variable, meaning that studies have been limited to dry dynamics only, where the model shows little evidence of sporadic convective behaviour."

6. Eqs. (1) "define the grad and nabla also explicitly please (it is 2D of course)": $\nabla$ has been defined in the following paragraph, "... and $\nabla = \begin{pmatrix} \partial/\partial x & \partial/\partial z \end{pmatrix}$ is the two-dimensional derivative."

7. After Eqs. (1) "C_o (subscript could be better)", "E_v (same)", "E_v -C_o": Notation suggestions (e.g. for $Co$ to be changed to $C_o$): thank you for the suggestion, but we would like to keep the current symbols if possible (the "o" is the second letter of "Condensation", rather than a qualifier symbol; similarly for $Ev$).

8. Sect. 2.3, start: "How is this done? Strange splitting?": It is now emphasised in the text that the micro-physics is still to be described, so it will not be evident at this stage. This kind of splitting is the usual way that numerical weather prediction models work.
"As explained above, Hydro-ABC solves equations (1) over a timestep $\Delta t$ in two stages, by first assuming $\mathcal{S}_{b'}$ and $Ev - Co$ are each zero (the dynamical core), and then their effects are added in a separate step (micro-physics, described below)."
We have also added the following to the earlier Sect. 2.2.2, before Eq. (4), "In its continuous form, Hydro-ABC is designed to conserve $\int \int dx dz \, E_{\mathrm{moist}}$ (see Appendix A). We, however, solve the Hydro-ABC equations (1) over a timestep $\Delta t$ in two parts: a 'dynamical core' in which $\mathcal{S}_{b'}$, $Ev$, and $Co$ are each zero, followed by a micro-physics step. A similar time-step splitting is done in other models, e.g. WRF (Weather Research and Forecasting model, Skamarock et al. (2021))."

9. Description around Eq. (5):

(a) "I am lost here: where is the expression for S_b'? These explanations are not sufficiently clear?" The text before and after Eq. (5) has been rewritten, which we hope describes a clearer path to Eq. (5). The derivation for $\mathcal{S}_{b'}$ is not made yet at this stage (comment is added to inform reader of this). There is no explicit equation for $\mathcal{S}_{b'}$, but how the value of $\mathcal{S}_{b'}$ is related to the solution of the equations is given in the last sentence of Sect. 2.3.

(b) "It seems that S_b' = -Lv(Ev-Co)/b' but that means that for b'=0 there is an issue unless of course Ev-Co scales with b' but I am missing that discussion (Ev-C0 scaling with b' means that essentially a L'Hopital rule of a kind can be/is used). Please clarify." This is a very interesting point. We find that the problem for $b' = 0$ is avoided by considering finite changes to $b'^2$ (instead of to $b'$), as made to get Eq. (7).

10. Equation (6): "Why is this approach convergent when Delta t goes to 0? The approach depends on the temporal resolution of the model, which is not desirable." We believe that the $(q_\mathrm{s} - q)\left(1 - e^{-\Delta t/\tau}\right)$ part of the parametrisation is reasonable given the following justification. Ignoring dynamics and changes in density one could approximate the equation for $q$ as $dq/dt = Ev - Co$. Assuming that $Ev - Co$ is proportional to $q_\mathrm{s} - q$, i.e. $Ev - Co = \tau^{-1}(q_\mathrm{s} - q)$, then this leaves one with the equation $dq/dt = \tau^{-1}(q_\mathrm{s} - q)$. Assuming that $q_\mathrm{s}$ is constant, the solution of this equation is $q(t) = q_\mathrm{s} + (q(0) - q_\mathrm{s})\exp(-t/\tau)$. Given this solution, and that the change in $q$ from 0 to $t$ is $q(t) - q(0)$, leads to $q(t) - q(0) = (q_\mathrm{s} - q(0))(1 - \exp(-t/\tau))$, which is the term in our parametrisation when $t = \Delta t$. This partially accounts for the fact that $\Delta t$ is finite.

11. Just before Eq. (7), "which equation (give reference to that eqn)": this has been changed to "According to (5), associated with this step is an increase in $b'^2$ by the amount $\Delta^\mathrm{mp}\left(b'^2\right) = 2A^2L_\mathrm{v}(q - q_\mathrm{s})$. This is due to the release of latent heat, which is converted to buoyancy energy. Writing $\Delta^\mathrm{mp}\left(b'^2\right)$ as $b'^2_\mathrm{mp} - b'^2$ (where $b'_\mathrm{mp}$ is the value of buoyancy after micro-physics, and $b'$ before), this equation for $\underline{\Delta^\mathrm{mp}\left(b'^2\right)}$ translates to"

12. Equation (7), "There seems to be a Delta t missing?": I think the equation given is correct (i.e. should not have a $\Delta t$), $\Delta^\mathrm{mp}\left(b'^2\right) = 2A^2L_\mathrm{v}(q - q_\mathrm{s})$ (the equation given in the text before (7)), comes from (5) ($\Delta^\mathrm{mp}\left(b'^2\right) = -2A^2L_\mathrm{v}\left(Ev - Co\right)\Delta t$), and $(Ev - Co)\Delta t$ is equal to $q - q_\mathrm{s}$, so the $\Delta t$ disappears there.

13. Section 2.3, point 1(a), "This step" has been changed to "Step (a)".

14. Section 2.3, point 1(b), "The treatment in (b) seems contrived; what is the underlying reason for the unphysical behaviour? Please address.": We believe that the unphysical effects arise due to the squaring of $b'$, which makes the sign of $b'$ invisible. The issue highlighted occurs only when $b'$ is negative. Since the primary aim of the model is to show some reasonable non-linear behaviour (rather than closely mimicking a real-world scenario), our documented 'solution' seemed satisfactory. There may be other solutions to this issue. The following sentence has been added to point 1(b): "The cause of this issue seems to be in the square (of $b'$) and square-root in Eq. (7), which forces the sign of $b'$ to become invisible, followed by the need to choose the sign of the solution."

15. Section 2.3, point 2, "no the needed". "the" has been removed. Also typo "an change" changed to "a change".

16. Section 2.3, point 2, "will become positive even instantly (maybe I don't understand)". Yes, this is correct, and this terminology has been weaved into the new text. "This process is restricted to when $b' > 0$, because in the case of $b' < 0$, (8) dictates that $b'$ will increase (and will instantly become positive). Even if one chooses to take the negative square-root in cases when $b' < 0$, (8) will yield an increased $b'$ (i.e. $b'_\mathrm{mp} - b' > 0$), which is unphysical."

17. Section 2.3, point 2, "However, the switches themselves are nonlinear functions." This point has been added to the text, "This avoids the need to solve the non-linear equations mentioned previously (but overall non-linearity is not avoided due the presence of the 'switches' in (6))."

18. Equation (9), "It needs "K" in 2 spots to be dimensionally correct. As well as "Pa"." Change has been made.

19. Section 2.5, "What time-step procedures are used, also with respect to the time splitting?" The new paragraph is as follows: "The dynamical core's scheme is based on (Cullen and Davies, 1991) and has two parts: an adjustment stage and an advection stage. The adjustment stage operates over two sub-timesteps and deals with the momentum and thermodynamic equations (omitting the advective terms) followed by treatment of the continuity equation. The advection stage advects the fields using the winds found in the adjustment stage averaged over the two sub-timesteps. Upwind gradients are used for the advection. For more details see Sect. 3.3 of Petrie et al. (2017). The equations for $q$ and $q_c$ (with $Ev - Co = 0$) are each solved by means of the same advection scheme as for the other variables in ABC. $Ev - Co$ is subsequently found with the micro-physics parametrisation of Sect. 2.3 using values of variables after the dynamical core is called."

20. Section 3, first paragraph, words added: "This involves extracting $u$ and $v$ from a longitude/height slice output from a 1.5km Unified Model (UM) run, which has the same grid staggering as the ABC model, and then adjusting $u$ and $v$ to have smooth periodic boundary conditions in $x$"

21. Section 3.1, L248, "same": we are not clear what the reviewer is suggesting here.

22. Section 3.1, L251, word added: "At this time, the second smaller plume has been extinguished by downdrafts from the main plume, but smaller convective plumes 11 to 13km have developed on either side of the main plume. Between 240 and 360 min, the smaller plumes are themselves impeded by down welling air on each side of the main plume."

23. Section 3.2: "Delta = 0.1 is used (just say what is without using "this setting")", change made: "$\Delta t = 0.1$s is the setting that has been used for the result in Fig. 2, and for the remainder of this paper."

24. Section 3.3, "delta z?": the author is asking about the dependence of the maximum wavenumber on $\Delta z$. We are referring to horizontal wavenumbers only, and so $\Delta z$ is not relevant. New text: "In practice though the maximum horizontal wavenumber and frequency are set by $\Delta x$ (the grid box length) and $\Delta t$ (the timestep) respectively."

25. Figure 3, "what causes the increase [in energy], as one could allow for a scheme that only decays. although I don't mind." This is not a question we can easily answer, but since the increase in energy is small, we are not concerned that it is a significant problem in practice. Extra sentence added to Section 3.2: "The slight increase in energy in the $\Delta t = 0.1$s and 0.2s integrations is small, but unexpected."

26. Section 3.3, L287:

    (a) "how do you know and how did you define these [low-wavenumber inertio-gravity waves]?" This interpretation is from the frequency of the features in Fig. 4. Please see the discussion in Sect. 3.3 up to the Gill (1982) reference. Note that the Lamb line was plotted incorrectly in the original manuscript, so now no gravity waves have a frequency higher than this line (see below).

    (b) "That is odd; why are these there [inertio-gravity waves have frequencies higher than the Lamb line]?", "Investigation of the linear model seems in order here." : A re-analysis of the linear system shows that we have plotted the Lamb line incorrectly. Figure 4 (see below) has been corrected and the comment about some low-wavenumber inertio-gravity waves have frequencies higher than the Lamb line has been replaced with the following, "In the (Hydro)-ABC runs some of the high-wavenumber acoustic waves have frequencies lower than the Lamb line (visible especially in the top-right section of the bottom-right panel). This is a numerical effect where the frequencies of high-wavenumber waves are too low, leading to numerical dispersion (see e.g. Chapter 2 of Durran, 1999)."

27. End of Sect. 3.3. "Double resolution runs would be relevant herein." We assume that this comment relates to the results of the dispersion curves in Fig. 4. We are not sure how our results justify/demand a need for higher resolution runs.

28. Section 4.1, "refer to eqn (9) too". New text: "Each ensemble member is prepared with the same $q$ and $q_c$ fields (as in Fig. 2), but the relative humidity ($q/q_s$, the field that matters for convection) is different between members because $q_s$ is a function of $b'$ and $\tilde{\rho}'$ (see Eq. (9) Sect. 2.4, where $p$ and $T$ depend on $\tilde{\rho}'$ and $b'$ as shown in Eqs. (10) and (12))."

29. Section 4.1, L356, "how do I see this and where?" (relating to the proposed characteristic of hydrostatic balance (HB) diagnosed from Fig. 6). The text has been re-written, "Thus the property of HB is assumed when the vertical derivative of the $\tilde{\rho}'$-$\tilde{\rho}'$ correlation function at a point in the top three rows of Fig. 6 is zero, and is matched by the $b'$-$\tilde{\rho}'$ correlation also being zero at the corresponding point in the bottom three panels. This property is seen at many points in Fig. 6 and a sample of points is highlighted below."

30. Section, L406-407, "typo": this has been corrected. The new text is, "The CAPE (red) increases towards the convection zone, . . . "

31. Figure 7, "CIN seems rather pointless.": CIN (convective inhibition) is considered because it shows the right behaviour. It may turn out to be more useful when we use it in future work. For example, a small value of CIN may turn out to be a reliable indicator of convection, or at least one of the required conditions.

32. Section 5, L425, "which is not clear" (the determination of $\mathcal{S}_{b'}$ and $Ev - Co$). We hope that this has been addressed in the revision of Sect. 2.3 (see previous points 8 - 17).

33. Section 5, L454: "I can't follow the last part of this sentence. Please change and clarify.". The new text is, "Many operational systems base their static background error covariance model, or the training data used to calibrate it, on hydrostatic balance (Berre, 2000; Bannister, 2008; Gustafsson et al., 2018; Bannister et al., 2020), and so this result may urge changes to this practice."

34. Section 5, L454-456, "I do not understand this sentence and statement either.", "What? Since previous sentence was mangled, the next sentence to this sentence is incomprehensible. Also avoid using the word This without qualifier." The new text is, "Given that the covariances depend on convection, it is necessary to be able to diagnose reliably the presence of convection (or impending convection) especially if a suitable ensemble is not available, and one needs to instead prescribe covariances for the DA. Such an indicator can be applied to the background state in a DA application, in order to decide the most relevant set of error covariances."

35. Section 5, L460, "I felt CIN was rather pointless and telling very little. What did I miss?" All we are saying here in the conclusions is that CIN (convective inhibition) was considered.

36. Appendix A, "So what is the mathematical expression resulting there from? Please supply." This is a request to give the expression for CAPE (convective available potential energy). This is done in the paper with the following "In the real atmosphere CAPE is proportional to the vertical integral of the difference between a moist parcel's temperature, $\hat{T}$, and the environmental temperature, $T_{\text{env}}$ (e.g. Sect. 7.4.1 of Salby (1996), $\text{CAPE} = g \int_{\text{LFC}}^{\text{LNB}} \frac{\hat{T} - T_{\text{env}}}{T_{\text{env}}} dz$, where LFC is the level of free convection and LNB is the level of neutral buoyancy $-$ see below)."

37. Appendix A, L467, "In Hydro-ABC there is no temperature variable, . . . ", "but there is; you derived one in (12).": The new text is, "In Hydro-ABC there is no explicit temperature variable, . . . "

38. Appendix A, L474-475, "what is the formal definition of environmental profile?". The new text is, "The environmental profile (the mean state around the convecting column, found by averaging the model fields over 11 points horizontally centred on the profile of interest) is supposed to be in hydrostatic equilibrium." (another change is made to this text in point 39 below).

39. Appendix A, L475, "but is it; where is it shown that it is? [That the environmental profile is in hydrostatic equilibrium]". This is a standard approximation for the environment. We have tested the hydrostatic imbalance of the environmental profile and found that when averaging over 101 points

horizontally (in the original version this was 11 points), the environmental profile has very small hydrostatic imbalance compared to the convecting column. The hydrostatic imbalance is shown in Fig. A below (not shown in the paper). Averaging over 101 point (rather than 11) does not change the CAPE and CIN plots noticeably. The new text (incorporating also point 38) is, "The environmental profile (the mean state around the convecting column, found by averaging the model fields over 101 points horizontally centred on the profile of interest) is  found to be in hydrostatic equilibrium (not shown):"

40. Appendix A, Eq. (A6), "Why is relationship 12 not used here? I don't understand how the definition of CAPE is related to what is calculated." Equation (12) is not used because it is an approximation to what temperature might look like in this system (it is based on an assumed pressure profile e.g.), so this might not be reliable. We believe it is better to seek a form of CAPE that is based directly on ABC's equations of motion, which is what is done in the appendix.

   (a) The following text has been added: "A positive value indicates energy is available for convection, and is thus a measure of instability. The above equation for CAPE [standard CAPE equation added involving temperature variable] is relevant to the standard equations of motion. In Hydro-ABC there is no explicit temperature variable, but we can derive a physically reasonable CAPE-like quantity by analysing the energetics of an air parcel using the ABC equations."

   (b) The following text has been added after Eq. (A7): "A large CAPE is associated with large positive differences $\hat{b}'(z) - b'_{\text{env}}(z)$ between $z_{\text{LFC}}$ and $z_{\text{LNB}}$, which is associated with buoyant air (air locally warmer than its surroundings). Similarly a large value of CIN is associated with the reverse – cooler air at a given location compared to its surroundings."

[Figure]

Figure 4: Vertical wind in the longitude/time (top panels) and wavenumber/frequency (bottom panels) domains for level 30 in the model (7.7km height). The left panels are for dry ABC and the right panels are for hydro-ABC. The initial conditions of $u$, $v$, $w$, $\tilde{\rho}'$, and $b'$ are identical for ABC and Hydro-ABC, and the additional initial conditions for $q$ and $q_c$ are the same as those used in Fig. 2. In the bottom panels, the horizontal dashed lines are each at frequency $A/(2\pi)$ (the pure gravity wave frequency), and the sloped dashed lines each has gradient $df/dn = L^{-1}\sqrt{BC}$ (the pure sound wave speed), where $f$ is frequency (the $y$-axes), $n$ is the wavenumber index (the $x$-axes), and $L$ is the length of the domain, $540 \times 10^3$m.

[Figure]

Figure 5: **Figure A** (not shown in the paper): Hydrostatic imbalance (HI) profiles at three locations at $t = 10\text{min}$. The first profile is in the convection zone (the position of the first vertical dashed line in Fig. 7), the second profile is in the moist zone (not shown in Fig. 7), and the third profile is also in the moist zone (the position of the second vertical dashed line in Fig. 7). Two lines are drawn for each point: the blue lines are the HI for the profile in question and the orange lines are the HI for the environment close to the point (average of 50 grid points either side of the convecting point (101 points in total)). Given a vertical profile (an actual profile or a horizontally averaged one), the HI is defined as $(C\partial\tilde{\rho}'/\partial z - b')\,/\,(\text{RMS}\,(C\partial\tilde{\rho}'/\partial z) + \text{RMS}\,(b'))$.

---

## Author Response (AR2)

Dear Prof. Chiel van Heerwaarden,

Tuesday 19th September 2023

Many thanks for accepting our publication, "The "Hydro-ABC model" (Vn 2.0): a simplified convective-scale model with moist dynamics" in Geoscientific Model Development.

No changes were requested for this version, but we have changed Fig. 6 as we found the original version to have failed colour-blind tests. Different shaped symbols are now used instead of different coloured symbols. The corresponding parts of the referring text have been changed accordingly.

Once again many thanks for considering our paper.

Yours sincerely,

Ross Bannister (on behalf of both authors)